# High-fidelity neural speech reconstruction through an efficient acoustic-linguistic dual-pathway framework

Jiawei Li[1,2], Chunxu Guo[1], Chao Zhang[3,4], Edward F Chang[5], Yuanning Li[1,2,4,6,7]*

[1]School of Biomedical Engineering, ShanghaiTech University, Shanghai, China; [2]State Key Laboratory of Advanced Medical Materials and Devices, ShanghaiTech University, Shanghai, China; [3]Department of Electronic Engineering, Tsinghua University, Beijing, China; [4]Shanghai Artificial Intelligence Laboratory, Shanghai, China; [5]Department of Neurological Surgery, University of California, San Francisco, San Francisco, United States; [6]Shanghai Clinical Research and Trial Center, Shanghai, China; [7]Lin Gang Laboratory, Shanghai, China

## eLife Assessment

This study presents a **valuable** advance in reconstructing naturalistic speech from intracranial ECoG data using a dual-pathway model. The evidence supporting the claims of the authors is **solid**. This work will be of interest to cognitive neuroscientists and computer scientists/engineers working on speech reconstruction from neural data.

*For correspondence:
yuanningli@gmail.com

**Abstract** Reconstructing speech from neural recordings is crucial for understanding human speech coding and developing brain-computer interfaces (BCIs). However, existing methods trade off acoustic richness (pitch, prosody) for linguistic intelligibility (words, phonemes). To overcome this limitation, we propose a dual-path framework to concurrently decode acoustic and linguistic representations. The acoustic pathway uses a long-short term memory (LSTM) decoder and a high-fidelity generative adversarial network (HiFi-GAN) to reconstruct spectrotemporal features. The linguistic pathway employs a transformer adaptor and text-to-speech (TTS) generator for word tokens. These two pathways merge via voice cloning to combine both acoustic and linguistic validity. Using only 20 min of electrocorticography (ECoG) data per human subject, our approach achieves highly intelligible synthesized speech (mean opinion score = 4.0/5.0, word error rate = 18.9%). Our dual-path framework reconstructs natural and intelligible speech from ECoG, resolving the acoustic-linguistic trade-off.

## Introduction

Understanding how the human brain encodes and decodes spoken language is a central challenge in cognitive neuroscience (*Hickok and Poeppel, 2007*; *Bhaya-Grossman and Chang, 2022*; *Huth et al., 2016*; *Hamilton et al., 2021*; *Mesgarani et al., 2014*), with profound implications for brain-computer interfaces (BCIs) (*Silva et al., 2024*; *Stavisky, 2025*; *Qiu et al., 2025*; *Conti, 2024*). Reconstructing perceived speech directly from neural activity provides a powerful approach to studying the hierarchical organization of language processing in the auditory cortex and to developing assistive technologies for individuals with impaired communication abilities (*Akbari et al., 2019*; *Guenther et al., 2009*; *Herff et al., 2015*; *Bellier et al., 2023*; *Willett et al., 2023*; *Pasley et al., 2012*). This neural decoding paradigm not only helps reveal the representational structure of linguistic features across

the cortex but also enables the synthesis of intelligible speech directly from brain signals (*Metzger et al., 2023*; *Wairagkar et al., 2025*; *Anumanchipalli et al., 2019*).

Recent advances in deep learning have accelerated progress in neural speech reconstruction by enabling end-to-end mappings from neural recordings to acoustic waveforms (*Silva et al., 2024*; *Stavisky, 2025*; *Akbari et al., 2019*; *Willett et al., 2023*; *Chen et al., 2024b*; *Komeiji et al., 2022*). However, these models typically require large-scale paired datasets, hours or even days of simultaneous neural and speech recordings per subject, to learn the complex nonlinear transformations involved. This data scarcity remains a major bottleneck, especially in clinical or intracranial recording settings where data acquisition is constrained by practical and ethical limitations.

To address this challenge, recent work has leveraged pre-trained self-supervised learning (SSL) models, such as Wav2Vec2.0 and HuBERT, which learn rich acoustic and phonetic representations from large-scale unlabeled speech corpora (*Baevski et al., 2020*; *Hsu et al., 2021*). These models have been shown to exhibit strong correspondence with neural activity in the auditory cortex, supporting near-linear mappings between latent model representations and cortical responses (*Millet et al., 2022*; *Li et al., 2023*; *Chen et al., 2024a*). Similarly, large language models trained purely on text, such as GPT (*Radford et al., 2019*), also capture high-level semantic representations that correlate with activity in language-responsive cortical regions (*Goldstein et al., 2022*; *Schrimpf et al., 2021*; *Tuckute et al., 2024*). Together, these findings suggest that the latent spaces of speech and language models provide a natural bridge between brain activity and interpretable representations of speech, enabling more data-efficient neural decoding.

Current approaches to speech decoding from brain activity can be broadly categorized into two paradigms. The first, neural-to-acoustic decoding, maps brain signals to low-level spectral or waveform features, such as formants, pitch, or spectrograms (*Akbari et al., 2019*; *Guenther et al., 2009*; *Pasley et al., 2012*; *He et al., 2025*). This approach captures fine acoustic detail, including speaker identity, prosody, and timbre, and benefits from pre-trained speech synthesizers for waveform generation (*Wairagkar et al., 2025*; *Anumanchipalli et al., 2019*; *Chen et al., 2024b*; *Li et al., 2024a*; *Liu et al., 2025*). However, due to the high dimensionality and dynamic nature of speech, this regression problem is inherently difficult and often yields low intelligibility when training data is limited. The second, neural-to-text decoding, treats speech reconstruction as a temporal classification task, mapping neural signals to a sequence of discrete linguistic units, such as phonemes, characters, or words (*Herff et al., 2015*; *Willett et al., 2023*; *Guo and Bhandari, 2025*; *Chen et al., 2025*; *Défossez et al., 2023*). This strategy achieves strong performance with relatively small datasets and aligns well with natural language processing (NLP) pipelines (*Duan et al., 2024*; *Tang et al., 2023*; *Feng et al., 2025*; *Zhang et al., 2024*), but sacrifices naturalness and expressiveness and lacks paralinguistic features crucial for human communication.

Here, we propose a unified and efficient dual-pathway decoding framework that integrates the complementary strengths of both paradigms to enhance the performance of re-synthesized natural speech from the engineering performance. Our method maps intracranial electrocorticography (ECoG) signals into the latent spaces of pre-trained speech and language models via two lightweight neural adaptors: an acoustic pathway, which captures low-level spectral features for naturalistic speech synthesis, and a linguistic pathway, which extracts high-level linguistic tokens for semantic intelligibility. These pathways are fused using a fine-tuned text-to-speech (TTS) generator with voice cloning, producing re-synthesized speech that retains both the acoustic spectrotemporal details, such as the speaker's timbre and prosody, and the message linguistic content. The adaptors rely on near-linear mappings and require only 20 min of neural data per participant for training, while the generative modules are pre-trained on large unlabeled corpora and require no neural supervision.

## Results
### Overview of the proposed speech re-synthesis method
Our proposed framework for reconstructing speech from intracranial neural recordings is designed around two complementary decoding pathways: an acoustic pathway focused on preserving low-level spectral and prosodic detail, and a linguistic pathway focused on decoding high-level textual and semantic content. For every participant, our adaptor is independently trained, and we select speech-responsive electrodes (selection details are provided in the Methods section) to tailor the model to

individual neural patterns. These two streams are ultimately fused to synthesize speech that is both natural-sounding and intelligible, capturing the full richness of spoken language. *Figure 1* provides a schematic overview of this dual-pathway architecture.

The acoustic pathway aims to synthesize speech with high acoustic naturalness, including prosody, speaker identity, and timbre, through a two-stage training process. In stage 1, we pre-train a high-fidelity generative adversarial network (HiFi-GAN) generator using a large corpus of natural speech data, without requiring paired neural recordings. This stage involves a speech autoencoder architecture consisting of a frozen Wav2Vec2.0 encoder (*Baevski et al., 2020*), which transforms ECoG signals into latent representations known to correlate with brain activity (*Millet et al., 2022*; *Li et al., 2023*), and a trainable HiFi-GAN decoder, which synthesizes natural-sounding speech waveforms from these representations (*Figure 1B*). HiFi-GAN (*Kong et al., 2020*), a generative adversarial network, was pre-trained to reconstruct high-fidelity speech using features extracted by the Wav2Vec2.0 encoder. This stage defines a rich, brain-aligned acoustic feature space (*Chen et al., 2024a*) that does not require paired neural data. In stage 2, in order to decode speech from the brain, we next trained a lightweight acoustic adaptor that maps ECoG recordings onto the pre-defined latent space of the speech generator. This adaptor, a three-layer bidirectional long-short term memory (LSTM) model (~9.4 M parameters), was trained using only 20 min of neural recordings per participant while listening to natural speech. The HiFi-GAN waveform decoder remained frozen, allowing the adaptor to be trained efficiently and independently. This acoustic pathway enables high-fidelity speech reconstruction directly from neural signals, capturing fine-grained auditory detail while remaining data-efficient.

While the acoustic pathway ensures naturalness, it lacks explicit linguistic structure. The goal of the linguistic pathway is to decode discrete word-level representations from brain activity, facilitating the reconstruction of speech that is not only fluent but also semantically and syntactically accurate. To extract high-level linguistic information, we trained a Transformer-based linguistic adaptor (~10.1 M parameters) to align ECoG signals with word token representations derived from transcribed speech. This adaptor comprises a three-layer encoder and decoder and is trained in a supervised manner using the same 20 min neural dataset. The adaptor output is fed into Parler-TTS (*Lyth and King, 2024*), a state-of-the-art TTS model (~880 M parameters), which synthesizes fluent speech from word token sequences. This pathway captures syntactic structure and lexical identity that are not explicitly represented in the acoustic stream.

To generate speech that combines the strengths of both pathways, retaining the acoustic richness from the first and the intelligibility from the second, we fused the outputs in a final synthesis stage. The resulting waveforms were passed through CosyVoice 2.0 (*Du et al., 2024*), a high-quality speech synthesis model with robust voice cloning and denoising capabilities. CosyVoice 2.0 was fine-tuned on the training set of TIMIT to improve articulation clarity, speaker-specific characteristics, and prosodic expressiveness. The final output preserves both low-level acoustic fidelity (e.g. pitch, prosody, and timbre) and high-level linguistic content (e.g. accurate word sequences), resolving the traditional trade-off in brain-to-speech decoding (*Figure 2A*). Across the entire framework, unsupervised pretraining and supervised neural adaptation are used synergistically to achieve efficient, high-quality speech reconstruction (*Figure 2—figure supplement 1*).

## The acoustic and linguistic performance of the reconstructed speech

We collected intracranial neural recordings from nine participants (4 male and 5 females; age range: 31–55) undergoing clinical neurosurgical evaluation for epilepsy. Each participant was implanted with high-density ECoG electrode arrays placed over the superior temporal gyrus and surrounding auditory speech cortex, based on clinical needs. During the experiment, participants passively listened to a set of 599 naturalistic English sentences spoken by multiple speakers. Neural signals were recorded at high temporal resolution while preserving the spatial specificity necessary to capture cortical responses to speech. This yielded approximately 20 min of labelled neural data for each participant. We then evaluated the quality of the re-synthesized sentences for each participant in the test set (*Figure 3*, *Figure 3—figure supplement 1*). Both subjective and objective evaluation metrics were adopted to give a comprehensive quantification of the effectiveness of this dual-path strategy in reconstructing intelligible and naturalistic speech from limited neural data (*Figure 2B*). This included mean opinion score (MOS), mel-spectrogram correlation, word error rate (WER), and phoneme error rate (PER).

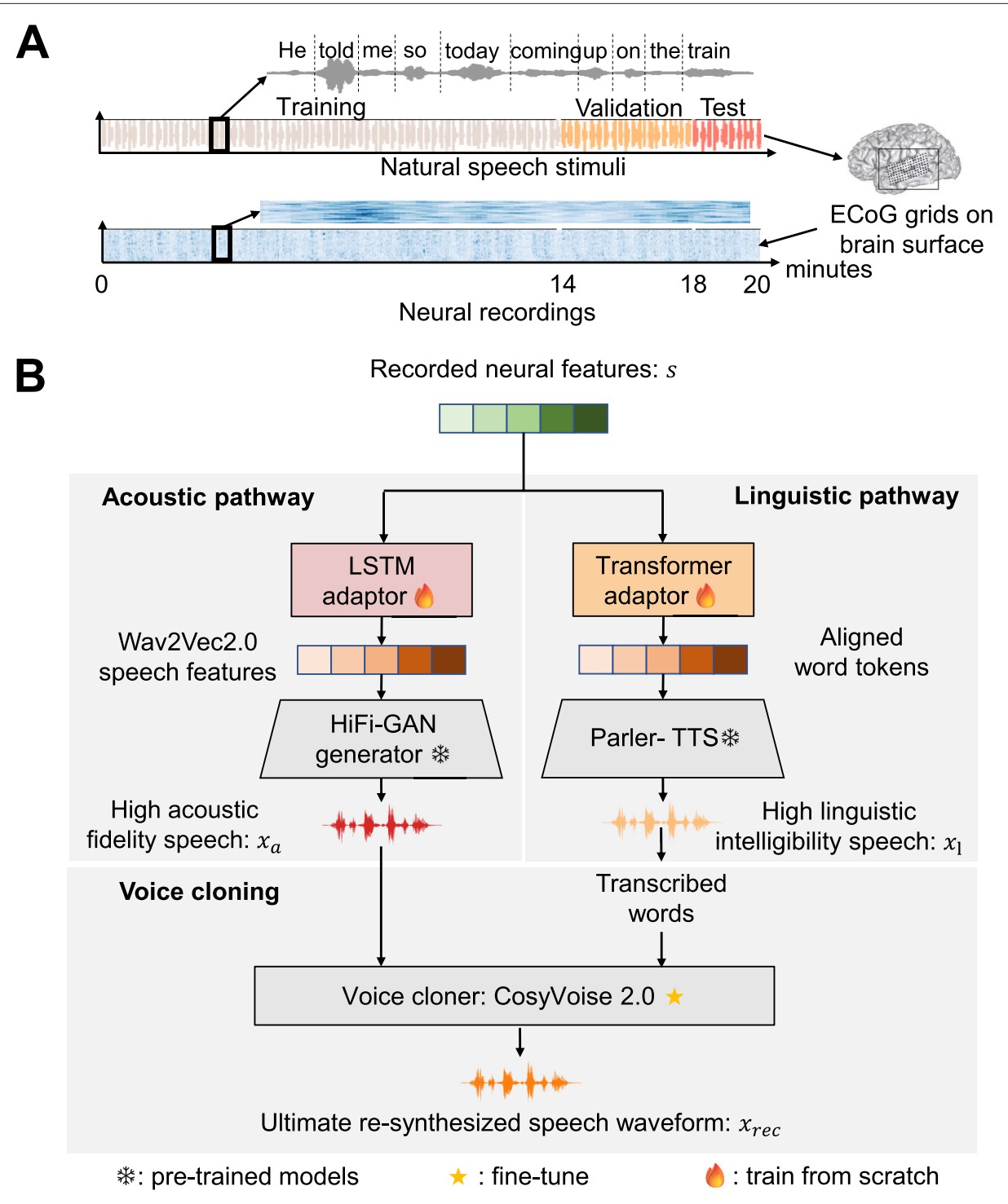

**Figure 1.** The neural data acquisition and acoustic-linguistic dual-pathway framework for neural-driven natural speech re-synthesis. (**A**) The neural data acquisition. We collected electrocorticography (ECoG) data from nine monolingual native English participants as they each listened to English sentences from the TIMIT corpus, resulting in 20 min of neural recording data per participant. Furthermore, for each participant, 70%, 20%, and 10% of the recorded neural activities are, respectively, allocated to the training, validation, and test sets randomly. (**B**) The acoustic-linguistic dual-pathway framework. The acoustic pathway consists of two stages: Stage 1: A high-fidelity generative adversarial network (HiFi-GAN) generator is pre-trained to synthesize natural speech from features extracted by the frozen Wav2Vec2.0 encoder, using a multi-receptive field fusion module and adversarial training with discriminators. This stage uses the LibriSpeech corpus to enhance speech representation learning. Stage 2: A lightweight long-short term memory (LSTM) adaptor maps neural activity to speech representations, enabling the frozen HiFi-GAN generator to re-synthesize high acoustic fidelity speech

*Figure 1 continued on next page*

*Figure 1 continued*

from neural data. The linguistic pathway involves a Transformer adaptor refining neural features to align with word tokens, which are fed into the frozen Parler-text-to-speech (TTS) model to generate high linguistic intelligibility speech. The voice cloning stage uses CosyVoice 2.0 (fine-tuned on TIMIT) to clone the speaker's voice, ensuring the ultimate re-synthesized speech waveform matches the original stimuli in clarity and voice characteristics.

The online version of this article includes the following figure supplement(s) for figure 1:

**Figure supplement 1.** Speech-responsive and tone-discriminating electrodes for all participants.

First, we evaluated the lower-level acoustic quality of the reconstructed speech. The average mel-spectrogram $R^2$ (*Figures 2B and 3B*) for the neural-driven re-synthesized speech across participants was 0.824±0.029 (mean ± s.e., the best performance across participants achieving 0.844±0.028, *Figure 3—figure supplement 1B*). For comparison, we also computed these metrics on surrogate speech with additive noise (*Xu et al., 2020*) (measured in dB). The quality of the reconstructed speech was comparable to the $R^2$ of –5 dB and 0 dB additive noise (–5 dB: mel-spectrogram $R^2$=0.864 ± 0.019, *p*=0.161, two-sided t-test; 0 dB: mel-spectrogram $R^2$=0.771 ± 0.014, *p*=0.064, two-sided t-test). The average MOS (*Figures 2C and 3C*) for the neural-driven re-synthesized speech across participants was 3.956±0.173, with the best performance across participants reaching 4.200±0.119 (*Figure 3—figure supplement 1C*).

Second, we further evaluated the linguistic content and intelligibility of the re-synthesized speech using word error rate (WER) and phoneme error rate (PER) metrics (*Figures 2D–E and 3D–E*). The neural-driven re-synthesized speech achieved an average WER of 18.9±3.3% across participants, with the best performance reaching 12.1±2.1% (*Figure 3—figure supplement 1D*). The WER result is comparable to adding –5 dB noise on the original input speech stimuli (14.0±2.1%, *p*=0.332, two-sided t-test). Similarly, the average PER was 12.0±2.5%, with the best performance reaching 8.3±1.5% (*Figure 3—figure supplement 1E*), comparable to adding –5 dB noise on the original input speech stimuli (6.0±1.5%, *p*=0.068, two-sided t-test).

To evaluate the data efficiency of our framework and its sensitivity to the quantity of available neural recordings, we performed an ablation study by training models on progressively smaller subsets of the full per-subject training data (25%, 50%, 75%, and 100%). As shown in *Figure 3—figure supplement 2*, all three key performance metrics, acoustic fidelity (mel-spectrogram $R^2$, *Figure 3—figure supplement 2A*), lexical accuracy (WER, *Figure 3—figure supplement 2B*), and phonetic precision (PER, *Figure 3—figure supplement 2C*), improved monotonically with increased training data volume.

## The acoustic pathway reconstructs high-fidelity lower-level acoustic information

The acoustic pathway, implemented through a bi-directional LSTM neural adaptor architecture (*Figure 1B*), specializes in reconstructing fundamental acoustic properties of speech. This module directly processes neural recordings to generate precise time-frequency representations, focusing on preserving speaker-specific spectral characteristics like formant structures, harmonic patterns, and spectral envelope details. Quantitative evaluation confirms its core competency: achieving a mel-spectrogram $R^2$ of 0.793±0.016 (*Figure 3B*) demonstrates remarkable fidelity in reconstructing acoustic microstructure. This performance level is statistically indistinguishable from original speech degraded by 0 dB additive noise (0.771±0.014, *p*=0.242, one-sided t-test). We chose a bidirectional LSTM architecture for this adaptor because its recurrent nature is particularly suited to modeling the fine-grained, short- to medium-range temporal dependencies (e.g. within and between phonemes and syllables) that are critical for acoustic fidelity. An ablation study comparing LSTM against Transformer-based adaptors for this task confirmed that LSTMs yielded superior mel-spectrogram reconstruction fidelity (higher $R^2$), as detailed in *Supplementary file 1*, likely by avoiding the over-smoothing of spectrotemporal details sometimes induced by the strong global context modeling of Transformers.

To confirm that the acoustic pathway's output is causally dependent on the neural signal rather than the generative prior of the HiFi-GAN, we performed a control analysis in which portions of the input ECoG recording were replaced with Gaussian noise. When either the first half, second half, or the entirety of the neural input was replaced by noise, the mel-spectrogram $R^2$ of the reconstructed speech dropped markedly, corresponding to the corrupted segment (*Figure 3—figure supplement 3*). This demonstrates that the reconstruction is temporally locked to the specific neural input and that

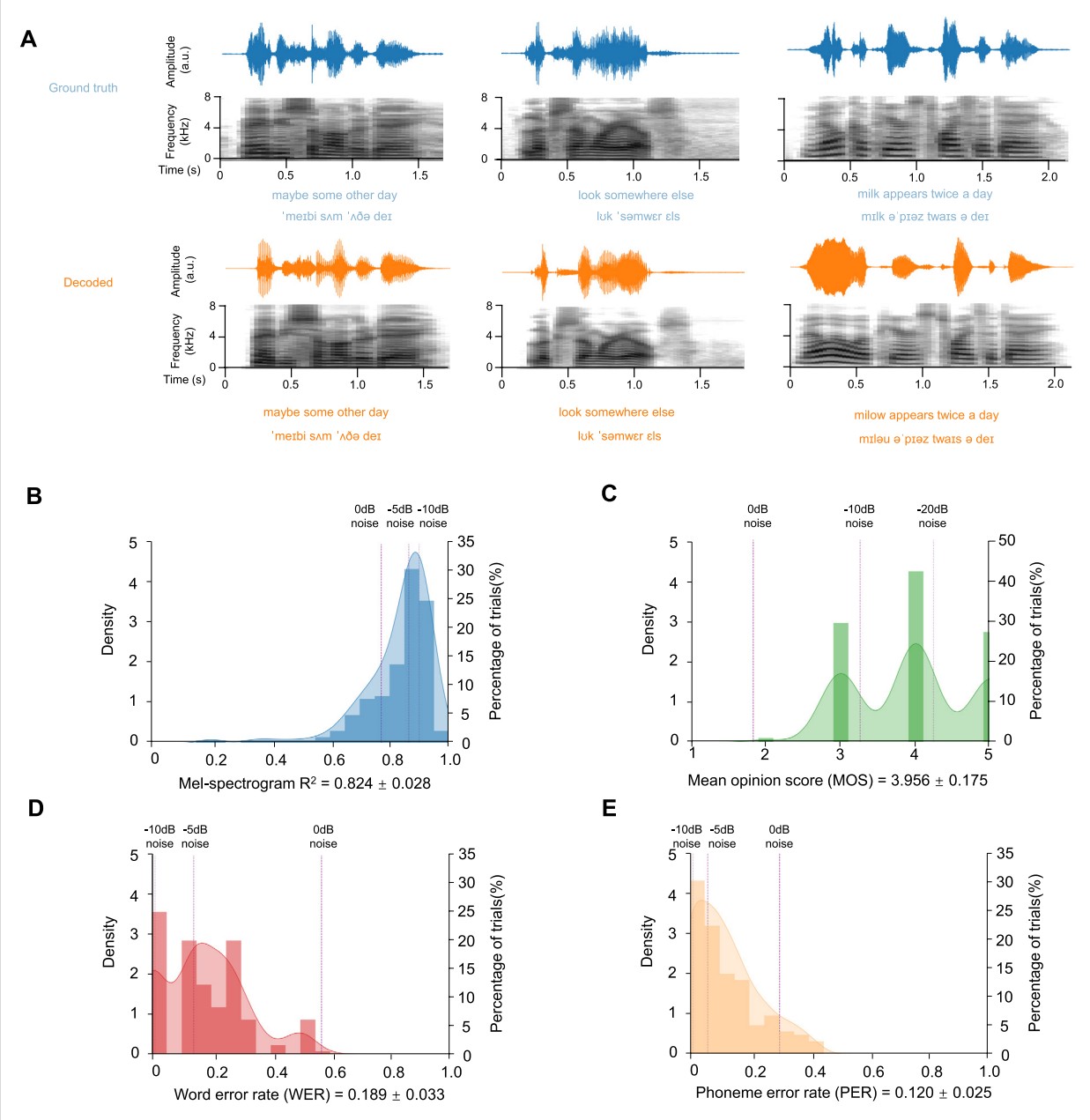

**Figure 2.** Neural-driven speech reconstruction performance. (**A**) Ground truth and decoded speech comparisons. Waveforms (time vs. amplitude) and mel-spectrograms (time vs. frequency range 0–8 kHz) are shown for both the original (top) and reconstructed (bottom) speech samples, demonstrating preserved spectral-temporal patterns in the neural-decoded output. Decoding demonstration of phonemes and words is attached to the speech. (**B**) Mel-spectrogram correlation analysis. The KDE curve shows the distribution of aligned correlation coefficients between the original and reconstructed mel-spectrograms across all test samples, while the bar chart represents the percentage of trials in the test set (mean = 0.824±0.028). Higher values reflect better acoustic feature preservation in the time-frequency domain. Purple dashed lines (light to dark) show average $R^2$ after adding –10 dB, –5 dB, and 0 dB additive noise to the original speech. (**C**) Human subject evaluations. The KDE curve displays the distribution of human evaluation results, while the bar chart represents the percentage of trials in the test set (mean = 3.956±0.175). Higher values reflect better intelligibility. Purple dashed lines (light to dark) show the average mean opinion score (MOS) after adding –20 dB, –10 dB, and 0 dB additive noise to the original speech. (**D–E**) Word Error Rate (WER) and Phoneme Error Rate (PER) assessment. evaluation. The KDE curve displays distribution, while the bar chart represents the percentage of trials in the test set (mean WER = 0.189±0.033, mean PER = 0.120±0.025). Lower values indicate better word-level reconstruction accuracy. Purple dashed lines (light to dark) show average WER after adding –10 dB, –5 dB, and 0 dB additive noise to the original speech.

The online version of this article includes the following figure supplement(s) for figure 2:

**Figure supplement 1.** An example of original and re-synthesized speech in all stages.

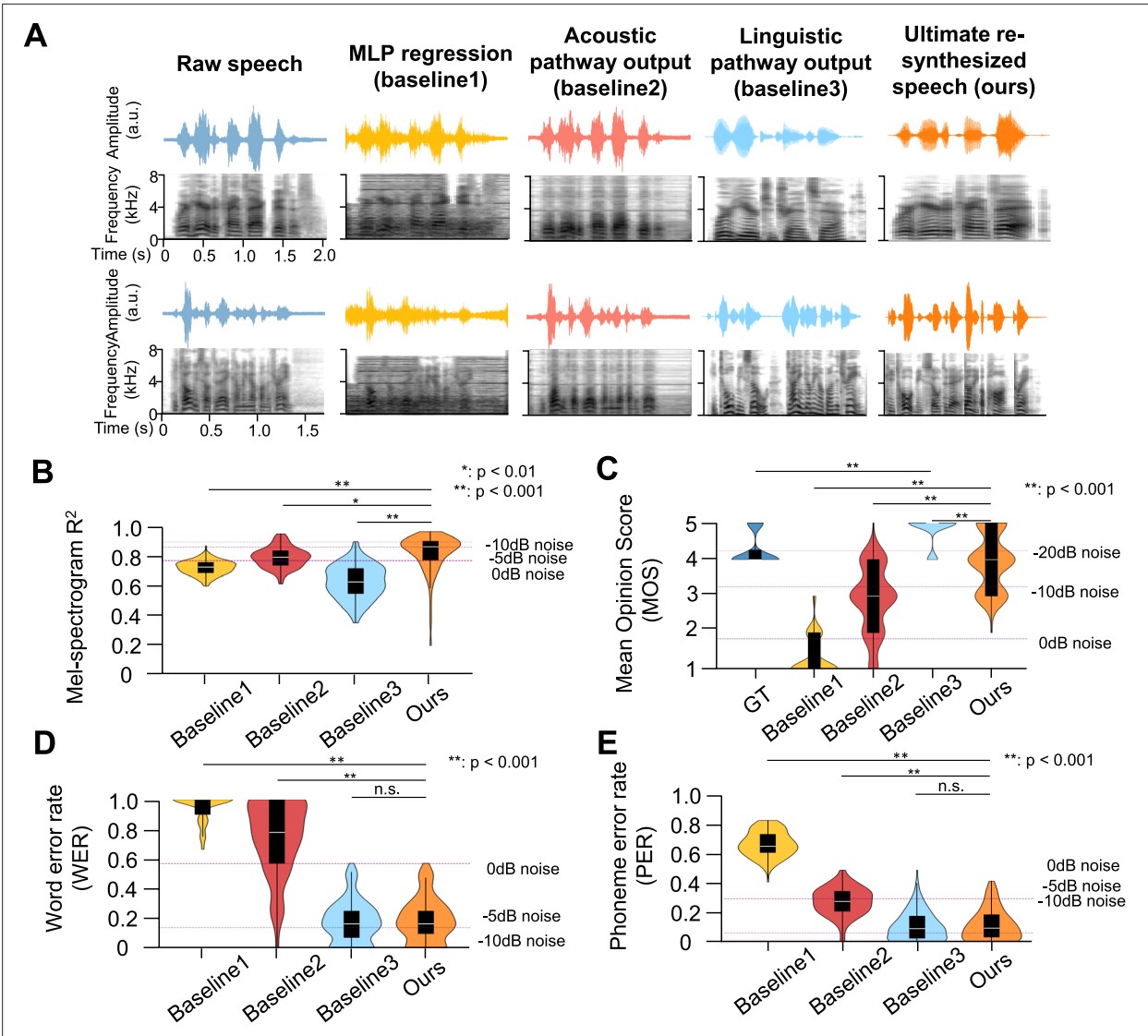

**Figure 3.** Performance comparison among re-synthesized speech and MLP regression, acoustic, and linguistic baselines. (**A**) Speech waveform and mel-spectrogram observations. Depicted are waveforms (time vs. amplitude) and mel-spectrograms (time vs. frequency ranging 0–8 kHz) for illustrative speech samples (Ground truth, MLP regression, and acoustic-linguistic pathway intermediate outputs as baseline 1–3, and our ultimate re-synthesized natural speech). (**B**) Objective evaluation using mel-spectrogram $R^2$. Violin plots show the aggregated distribution of $R^2$ scores (0–1 scale) assessing spectral fidelity. The white dot represents the median, the box spans the interquartile range, and whiskers extend to ±1.5× IQR. Three dashed lines (light to dark) show average $R^2$ after adding –10 dB, –5 dB, and 0 dB additive noise to the original speech. (**C**) Subjective quality evaluation using mean opinion score (MOS). Violin plots show aggregated MOS distribution (1–5 scale). Three dashed lines (light to dark) indicate average MOS after adding –20 dB, –10 dB, and 0 dB additive noise. Note: The higher MOS for Baseline 3 (linguistic pathway) compared to ground truth is due to the superior acoustic quality of the modern speech corpus used to train Parler-text-to-speech (TTS), whereas the TIMIT corpus contains inherent noise. GT: Ground truth. (**D**) Intelligibility assessment using word error rate (WER). Violin plots show aggregated WER distribution (0–1 scale). Three purple dashed lines (light to dark) show the average WERs after adding –10 dB, –5 dB, and 0 dB additive noise. (**E**) Phoneme error rate (PER) assessment. Format and noise conditions identical to panel **D**. Statistical significance markers: *$p$<0.01, **$p$<0.001, n.s.: not significant.

The online version of this article includes the following figure supplement(s) for figure 3:

**Figure supplement 1.** Performance comparison between re-synthesized speech and baselines for each participant.

**Figure supplement 2.** Model performance scales with the amount of training data.

**Figure supplement 3.** Control analysis: Sensitivity of the acoustic pathway's reconstruction to neural input.

the model does not 'hallucinate' spectrotemporal structure from noise. These results validate that the acoustic pathway performs genuine, input-sensitive neural decoding.

However, exclusive reliance on acoustic reconstruction reveals fundamental limitations. Despite excellent spectral fidelity, the pathway produces critically impaired linguistic intelligibility. At the word level, intelligibility remains unacceptably low (WER = 74.6±5.5%, *Figure 3D*), while MOS and phoneme-level precision fare only marginally better (MOS = 2.878±0.205, *Figure 3C*; PER = 28.1±2.2%, *Figure 3E*). These deficiencies manifest perceptually as distorted articulation, inconsistent prosody, and compromised phonetic distinctiveness, collectively rendering the output functionally unintelligible for practical communication. This performance gap underscores a fundamental constraint: accurate acoustic feature reconstruction alone is insufficient for generating comprehensible speech, necessitating complementary linguistic processing.

## The linguistic pathway reconstructs high-intelligibility, higher-level linguistic information

The linguistic pathway, instantiated through a pre-trained TTS generator (*Figure 1B*), excels in reconstructing abstract linguistic representations. This module operates at the phonological and lexical levels, converting discrete word tokens into continuous speech signals while preserving prosodic contours, syllable boundaries, and phonetic sequences. It achieves a mean opinion score of 4.822±0.086 (*Figure 3C*) - significantly surpassing even the original human speech (4.234±0.097, $p=6.674×10^{-33}$) in that the TIMIT corpus, recorded decades ago, contains inherent acoustic noise and relatively lower fidelity compared to modern speech corpus. Complementing this perceptual quality, objective intelligibility metrics confirm outstanding performance: WER reaches 17.7±3.2%, with PER at 11.0±2.3%. These results validate the pathway's capacity for reconstructing high-level linguistic structures from symbolic representations. We employed a Transformer-based Seq2Seq architecture for this adaptor to effectively capture the long-range contextual dependencies across a sentence, which are essential for resolving lexical ambiguity and syntactic structure during word token decoding. This choice was validated by an ablation study (*Supplementary file 2*), indicating that the Transformer adaptor outperformed an LSTM-based counterpart in word prediction accuracy.

Nevertheless, this linguistic feature caption comes at an acoustic cost. The pathway exhibits significant limitations in spectral fidelity, achieving only 0.627±0.026 (*Figure 3B*) mel-spectrogram $R^2$ - substantially below the 0 dB noise reference (0.771±0.014, $p=2.919 × 10^{-8}$). This deficiency stems from fundamental architectural constraints: lacking explicit acoustic feature modelling, the system fails to preserve critical speaker-specific attributes like vocal tract resonances, glottal source characteristics, and individual articulatory patterns. Consequently, while linguistically intelligible, the output lacks authentic speaker identity and natural timbral qualities essential for realistic speech reconstruction.

## The fine-tuned voice cloner further enhances the fidelity and intelligibility of re-synthesized speech

Voice cloning is used to bridge the gap between acoustic fidelity and linguistic intelligibility in speech reconstruction. This approach strategically combines the strengths of complementary pathways: the acoustic pathway preserves speaker-specific spectral characteristics while the linguistic pathway maintains lexical and phonetic precision. By integrating these components through neural voice cloning, we achieve balanced reconstruction that overcomes the limitations inherent in isolated systems. CosyVoice 2.0, the voice cloner module, serves specifically as a voice cloning and fusion engine, requiring two inputs: (1) a voice reference speech (provided by the denoised output of the acoustic pathway) to specify the target speaker's identity, and (2) a target word sequence (transcribed from the output of the linguistic pathway) to specify the linguistic content. The standalone baseline outputs of the two pathways can be integrated in this way.

The integration paradigm resolves this fundamental dichotomy through strategic combination. Where the acoustic pathway preserves spectral patterns ($R^2$=0.793±0.016) but suffers unintelligibility (WER = 74.6±5.5%), and the linguistic pathway delivers lexical precision but lacks acoustic fidelity, their fusion creates a complementary system. This synergistic architecture maintains the linguistic pathway's exceptional intelligibility while incorporating the acoustic pathway's spectrotemporal reconstruction. The resultant output achieves what neither can accomplish independently: linguistically precise

speech with an authentic acoustic signature, effectively decoupling and optimizing both reconstruction dimensions within a unified framework.

To validate this integrated approach, we established three baseline models for comprehensive benchmarking. Baseline 1 (MLP regression) directly maps neural recordings to mel-spectrograms without specialized processing. Baseline 2 represents the output of our acoustic pathway implementation, optimized for spectral feature reconstruction. Baseline 3 corresponds to the linguistic pathway output, generated by the TTS system without speaker-specific timbre adaptation.

The integration breakthrough occurs when combining this acoustic foundation with linguistic pathway components. The full pipeline achieves transformative improvements: WER decreases by 75% (18.9±3.3% vs 74.6±5.5%, $p=4.249×10^{-88}$, one-sided t-test, *Figure 2D*), while PER drops by 58% (12.0±2.5% vs 28.1±2.2%, $p=2.592×10^{-44}$, one-sided t-test, *Figure 2E*). Crucially, this intelligibility enhancement occurs without compromising acoustic fidelity: Mel spectrum $R^2$=0.824±0.029 vs 0.793±0.016 (acoustic pathway, $p=4.486×10^{-3}$, one-sided t-test).

To further elucidate the phoneme recognition performance, we conducted an in-depth analysis of the errors present in the transcribed phoneme sequences derived from both original and re-synthesized speech. Specifically, we focused on the re-synthesized speech, where individual characters were subjected to insertion, deletion, or substitution operations to align them with the ground truth. This process not only quantified the errors but also provided insights into the relationships between different phonemes.

The confusion matrices (*Figure 4A–D*) illustrate the distribution of these errors. The diagonal elements represent the correct matches, indicating the accuracy for each phoneme, while off-diagonal non-zero elements denote substitution errors. Empty rows signify insertion errors (extraneous phonemes), and empty columns indicate deletion errors (missing phonemes). These visual representations highlight the specific challenges faced by our proposed model and the baseline models in accurately recognizing certain phonemes.

In addition to the confusion matrices, we categorized the phonemes into vowels and consonants to assess the phoneme class clarity. We defined 'phoneme class clarity (PCC)' as the proportion of errors where a phoneme was misclassified within the same class versus being misclassified into a different class. The purpose of introducing PCC is to demonstrate that most of the misidentified phonemes belong to the same category (confusion between vowels or consonants), rather than directly comparing the absolute accuracy of phoneme recognition. For instance, a vowel being mistaken for another vowel would be considered a within-class error, whereas a vowel being mistaken for a consonant would be classified as a between-class error.

Our analysis revealed that the re-synthesized speech achieved a phoneme class clarity of 2.462±0.201 (*Figure 4E*), significantly outperformed chance level (1.360, calculated under the assumption that substituted phonemes are randomly distributed among all other phonemes, $p=3.424×10^{-6}$, one-sided t-test), MLP regression and acoustic pathway output (MLP regression: PCC = 1.694±0.134, $p=1.312×10^{-3}$, one-sided t-test; acoustic pathway output: PCC = 1.735±0.140, $p=2.367×10^{-3}$; linguistic pathway output: PCC = 2.446±0.203, $p=0.477$, one-sided t-test), indicating that most errors occurred within the same phoneme class. This result suggests that the model has a relatively good understanding of the phonetic structure, as it tends to confuse similar sounds rather than completely unrelated ones. Notably, this metric did not show significant differences compared to the baselines, suggesting that all models exhibit similar tendencies in terms of phoneme class confusion.

These comprehensive results confirm that our voice cloning framework successfully integrates the complementary strengths of acoustic and linguistic processing pathways. It achieves this integration without compromising either dimension: while significantly enhancing acoustic fidelity beyond what the linguistic pathway alone can achieve, it simultaneously maintains near-optimal linguistic intelligibility that the isolated acoustic pathway cannot provide. This balanced performance profile demonstrates the efficacy of our approach in producing high-quality, intelligible speaker-specific speech reconstruction.

## Discussion

In this study, we present a dual-path framework that synergistically decodes both acoustic and linguistic speech representations from ECoG signals, followed by a fine-tuned zero-shot text-to-speech network to re-synthesize natural speech with unprecedented fidelity and intelligibility.

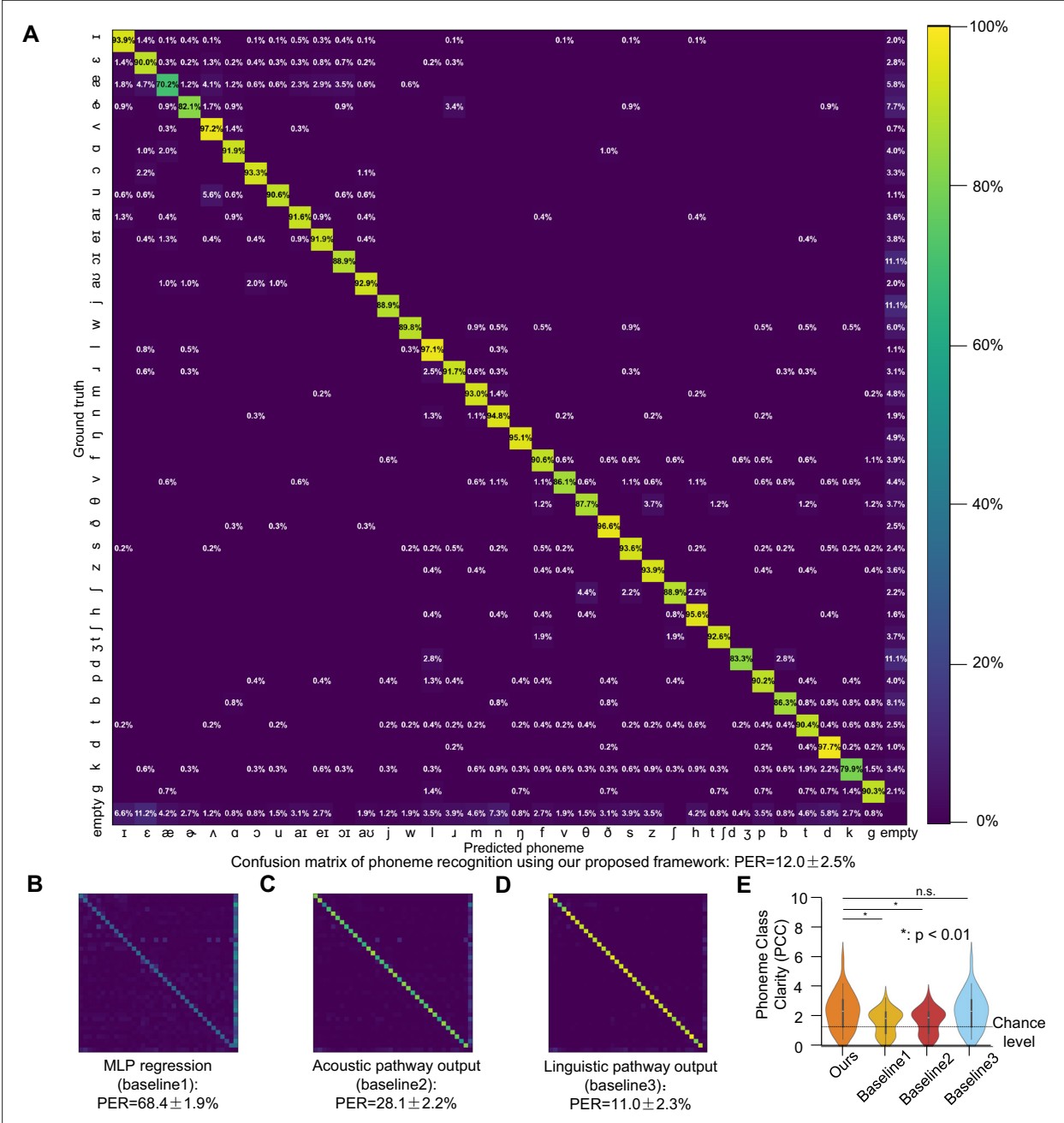

**Figure 4.** Phoneme recognition performance: confusion matrices and accuracy comparison. (**A**) Confusion matrix between transcribed phoneme sequences using our proposed model (horizontal axis) and ground truth (vertical axis). Diagonal values represent recognition accuracy for each phoneme (correct matches), while off-diagonal non-zero elements indicate substitution errors. Empty rows denote insertion errors (extraneous phonemes), empty columns indicate deletion errors (missing phonemes). All non-zero elements are highlighted for visual clarity. (**B–D**) Phoneme confusion matrix using baseline 1–3 for phoneme transcription. (**E**) Phoneme class clarity (PCC) across methods. PCC measures the proportion of mis-decoded phonemes that are confused within the same class (vowel-vowel or consonant-consonant) versus across classes (vowel-consonant). A higher PCC indicates that errors tend to be phonologically similar sounds, which supports intelligibility. The comparable PCC between our final integrated model and Baseline 3 (linguistic pathway) suggests that the phoneme-level error structure of our output is largely inherited from the high-quality linguistic prior embedded in the pre-trained text-to-speech (TTS) model (Parler-TTS). The violin plots show the distribution of PCC values across test samples. Asterisks (*) indicate phonemes where our proposed framework significantly outperforms baselines.

Crucially, by integrating large pre-trained generative models into our acoustic reconstruction pipeline and applying voice cloning technology, our approach preserves acoustic richness while significantly enhancing linguistic intelligibility beyond conventional methods. Our dual-pathway architecture, while inspired by converging neuroscience insights on speech and language perception, was principally designed and validated as an engineering solution. The primary goal is to build a practical decoder that achieves state-of-the-art reconstruction quality with minimal data. The framework's success is, therefore, ultimately judged by its performance metrics, high intelligibility (WER, PER), acoustic fidelity (mel-spectrogram $R^2$), and perceptual quality (MOS), which directly address the core engineering challenge we set out to solve. Using merely 20 min of ECoG recordings, our model achieved superior performance with a WER of 18.9±3.3% and PER of 12.0±2.5% (*Figure 2D and E*). This integrated architecture, combining pre-trained acoustic (Wav2Vec2.0 and HiFi-GAN) and linguistic (Parler-TTS) models through lightweight neural adaptors, enables efficient mapping of ECoG signals to dual latent spaces. Such methodology substantially reduces the need for extensive neural training data while achieving breakthrough word clarity under severe data constraints. The results demonstrate the feasibility of transferring the knowledge embedded in speech-data pre-trained artificial intelligence (AI) models into neural signal decoding, paving the way for more advanced brain-computer interfaces and neuroprosthetics.

Our framework establishes a framework for speech decoding by outperforming prior acoustic-only or linguistic-only approaches (*Supplementary file 3*) through integrated pretraining-powered acoustic and linguistic decoding. This dual-path methodology proves particularly effective where traditional methods fail, as it simultaneously resolves the acoustic-linguistic trade-off that has long constrained reconstruction quality. While end-to-end re-synthesis remains intuitively appealing, prior work confirms that direct methods achieve only modest performance given neural data scarcity (*Pasley et al., 2012*). To overcome this, we propose a hybrid encoder-decoder architecture utilizing: (1) pre-trained spectral synthesizers (Wav2Vec2.0 and HiFi-GAN) for acoustic fidelity, and (2) transformer-based token decoders (Parler-TTS) for linguistic precision. Participant-specific projection modules further ensure cross-subject transferability with minimal per-user calibration. Collectively, these advances surmount core limitations of direct decoding, enabling unprecedented speech quality within extreme data constraints.

A pivotal advancement lies in establishing robust, clinically relevant intelligibility metrics, notably achieving 18.9% WER and 12.0% PER through standardized evaluation, directly addressing the core challenge of word recognition in speech decoding. Our dual-path framework enables comprehensive sentence-level assessment through objective benchmarks (phonetic precision via PER and lexical accuracy via WER evaluated by Whisper ASR), acoustic fidelity validation (high mel-spectrogram correlation: 0.824±0.029, *Figure 2B*) and human perceptual testing (near-'excellent' MOS ratings: 3.956±0.173, *Figure 2C*). Critically, this tripartite evaluation spanning acoustic (spectral/time-domain), linguistic (phoneme/word), and perceptual dimensions revealed superior reconstruction quality, while objective metrics confirmed spectral coherence rivalling clean speech inputs.

The phoneme confusion pattern observed in our model output (*Figure 4A*) differs from classic human auditory confusion matrices. We attribute this divergence primarily to the influence of the Parler-TTS model, which serves as a strong linguistic prior in our pipeline. This component is trained to generate canonical speech from text tokens. When the upstream neural decoding produces an ambiguous or erroneous token sequence, the TTS model's internal language model likely performs an implicit 'error correction,' favoring linguistically probable words and pronunciations. This underscores that our model's errors arise from a complex interaction between neural decoding fidelity and the generative biases of the synthesis stage.

Our findings demonstrate that neural representations exhibit dual alignment, which is not only with acoustic features from deep speech models (Wav2Vec2.0), but critically with linguistic features from language models (Parler-TTS), establishing a bidirectional bridge between cortical activity and hierarchical speech representations. This convergence mirrors the complementary processing streams observed in the human speech cortex (*Li et al., 2023*; *Schrimpf et al., 2021*) where self-supervised models capture both spectrotemporal patterns and semantic structures. Such unified alignment marks a paradigm shift in brain-AI integration: The discovery of near-linear mappings between neural and multimodal AI spaces unlocks transformative applications in speech synthesis and cognitive interfaces. Furthermore, our results confirm that foundation models serve as scalable 'cognitive mirrors' (*Li et al.,*

*2024b*). With the advent of more sophisticated generative models, we anticipate enhanced neural decoding capabilities, including the potential to improve signal quality (in terms of SNR) and refine the generative model itself. Additionally, this framework extends beyond speech to other modalities, such as vision, suggesting that similar principles may apply to the generation of visual content from neural signals.

This work advances speech decoding and potentially speech BCIs by enabling real-time speech reconstruction with minimal neural data, improving scalability and practicality through pre-trained AI models, and suggesting potential for broader applications in perception, memory, and emotion. By adapting neural activity to a common latent space of the pre-trained speech autoencoder, our framework significantly reduces the quantity of neural data required for effective speech decoding, thereby addressing a major limitation in current BCI technologies. This reduction in data needs paves the way for more accessible and scalable BCI solutions, particularly for individuals with speech impairments who stand to benefit from immediate and intelligible speech reconstruction. Furthermore, the applicability of our model extends beyond speech, hinting at the possibility of decoding other cognitive functions once the corresponding neural correlates are identified. This opens up exciting avenues for expanding BCI functionality into areas, such as perception, memory, and emotional expression, thereby enhancing the overall quality of life for users of neuroprosthetic devices.

There are several limitations in our study. The quality of the re-synthesized speech heavily relies on the performance of the generative model, indicating that future work should focus on refining and enhancing these models. Currently, our study utilized English speech sentences as input stimuli, and the performance of the system in other languages remains to be evaluated. Regarding signal modality and experimental methods, the clinical setting restricts us to collecting data during brief periods of awake neurosurgeries, which limits the amount of usable neural activity recordings. Overcoming this time constraint could facilitate the acquisition of larger datasets, thereby contributing to the re-synthesis of higher-quality natural speech. Furthermore, the inference speed of the current pipeline presents a challenge for real-time applications. On our hardware (a single NVIDIA GeForce RTX 3090 GPU), synthesizing speech from neural data takes approximately two to three times longer than the duration of the target speech segment itself. This latency is primarily attributed to the sequential processing in the autoregressive linguistic adaptor and the computationally intensive high-fidelity waveform generation in the vocoder (CosyVoice 2.0). While the current study focuses on offline reconstruction accuracy, achieving real-time or faster-than-real-time inference is a critical engineering goal for viable speech BCI prosthetics. Future work must, therefore, prioritize architectural optimizations, such as exploring non-autoregressive decoding strategies and more efficient neural vocoders, alongside potential hardware acceleration. Additionally, exploring non-invasive methods represents another frontier; with the accumulation of more data and the development of more powerful generative models, it may become feasible to achieve effective non-invasive neural decoding for speech re-synthesis. Moreover, while our framework adopts specialized architectures (LSTM and Transformer) for distinct decoding tasks, an alternative approach is to employ a unified multimodal large language model (LLM) capable of joint acoustic-linguistic processing. Finally, the current framework requires training participant-specific adaptors, which limits its immediate applicability for new users. A critical next step is to develop methods that learn a shared, cross-participant neural feature encoder, for instance, by applying contrastive or self-supervised learning techniques to larger aggregated ECoG datasets. Such an encoder could extract subject-invariant neural representations of speech, serving as a robust initialization before lightweight, personalized fine-tuning. This approach would dramatically reduce the amount of per-subject calibration data and time required, enhancing the practicality and scalability of the decoding framework for real-world BCI applications.

In summary, our dual-path framework achieves high speech reconstruction quality by strategically integrating language models for lexical precision and voice cloning for vocal identity preservation, yielding a 37.4% improvement in MOS scores over conventional methods. This approach enables high-fidelity, sentence-level speech synthesis directly from cortical recordings while maintaining speaker-specific vocal characteristics. Despite current constraints in generative model dependency and intraoperative data collection, our work establishes a new foundation for neural decoding development. Future efforts should prioritize: (1) refining few-shot adaptation techniques, (2) developing non-invasive implementations, (3) expanding to dynamic dialogue contexts, and (4) cross-subject applications. The convergence of neurophysiological data with multimodal foundation models promises

transformative advances, not only revolutionizing speech BCIs but potentially extending to cognitive prosthetics for memory augmentation and emotional communication. Ultimately, this paradigm will deepen our understanding of neural speech processing while creating clinically viable communication solutions for those with severe speech impairments.

## Materials and methods

### Participants and ethics statement

The study comprised nine monolingual right-handed participants (4 male and 5 females; age range: 31–55), each with electrodes placed on the left hemisphere cortex to clinically monitor seizure activities. Electrode placement followed specific clinical requirements (see *Figure 1—figure supplement 1* for grid placement in each placement). All protocols in the current study were approved by the Institutional Review Board (IRB) of the University of California, San Francisco. Participants were thoroughly informed about the experiment and research and gave written consent. They volunteered for participation, ensuring that their involvement did not influence their clinical care. Additionally, verbal consent was obtained from participants at the beginning of each experimental session.

### Data acquisition and neural signal processing

All participants utilized high-density ECoG grids of uniform specifications and type. During the experimental tasks, neural signals captured by the ECoG grids were acquired using a multi-channel amplifier optically linked to the digital signal processor. Data recording was performed through Tucker-Davis Technology (TDT) OpenEx software. The local field potential at each electrode contact was amplified and sampled at 3052 Hz. Subsequent to data collection, the experiment employed the Hilbert transform to compute the analytic amplitudes of eight Gaussian filters (with center frequencies ranging from 70 to 150 Hz), and the signal was down-sampled to 50 Hz. These tasks were segmented into record blocks lasting approximately 5 min each. The neural signals were z-score normalized for each recording block.

### Experimental stimuli

The acoustic stimuli employed in this study comprised natural and continuous English speech. The English speech stimuli were sourced from the TIMIT corpus (*Garofolo et al., 1993*), consisting of 499 sentences read by 286 male and 116 female speakers. A silent interval of 0.4 s was maintained between sentences. The task was organized into five blocks, each with a duration of approximately 5 min.

### Electrode localization

In cases involving chronic monitoring, the electrode placement procedure involved pre-implantation MRI and post-implantation CT scans. In awake cases, temporary high-density electrode grids were utilized to capture cortical local potentials, with their positions recorded using the Medtronic neuro-navigation system. The recorded positions were then aligned to pre-surgery MRI data, with intraoperative photographs serving as additional references. The localization of the remaining electrodes was achieved through interpolation and extrapolation techniques.

### Speech-responsive electrode selection

The selection of speech-responsive electrodes played a crucial role in the re-synthesis of high-quality natural speech and in avoiding overfitting. Onsets were identified as the initiation of speech and are preceded by more than 400ms of silence. A paired sample t-test was conducted to investigate whether the average post-onset response (400 ms to 600 ms) exhibited a significant increase compared to the average pre-onset response (–200 ms to 0 ms, $p<0.01$, one-sided, Bonferroni corrected).

### Architecture and training of the speech Autoencoder

The speech Autoencoder served as a pivotal component, taking natural speech as input and reproducing the input itself. Within this model, the intermediate encoding layers were construed as the feature extraction layers. The architecture comprises an encoder, namely Wav2Vec2.0, and a decoder, which is the HiFi-GAN generator.

The Wav2Vec2.0 encoder was comprised of 7 1D convolutional layers that down-sample 16 kHz natural speech to 50 Hz, thereby extracting 768-dimensional local features. Additionally, it integrated 12 transformer-encoder blocks to extract contextual feature representations from unlabeled speech data.

On the other hand, the HiFi-GAN generator incorporated seven transposed convolutional layers for up-sampling, along with multi-receptive field (MRF) fusion modules. Each MRF module aggregated the output features from multiple ResBlocks. The discriminator within the HiFi-GAN encompassed both multi-scale and multi-period discriminators. Notably, the up-sample rates and kernel sizes of each ResBlock in the MRFs were meticulously configured. The up-sample rates of each ResBlock in the MRFs were set as 2, 2, 2, 2, 2, 2, and 5, respectively. The up-sample kernel sizes of each ResBlock in the MRFs were set as 2, 2, 3, 3, 3, 3, and 10. The periods of the multi-period discriminator were set as 2, 3, 5, 7, and 11.

The HiFi-GAN generator $G$ and discriminators $D_k$ were trained simultaneously and adversarially. The loss function consisted of three parts: adversarial loss $L_{Adv}$ to train the HiFi-GAN, mel-spectrogram loss $L_{Mel}$ to improve the training efficiency of the generator and feature matching loss $L_{FM}$ to measure the similarity between original speech and re-synthesized speech from learned features:

$$L_G = \sum_{k=1}^{K} \left[ L_{Adv}\left(G; D_k\right) + \lambda_{FM} L_{FM}\left(G, D_k\right) \right] + \lambda_{Mel} L_{Mel}\left(G\right)$$

The regularization coefficients $\lambda_{FM}$ and $\lambda_{Mel}$ were set as 2 and 50, respectively in the implementation. In detail, the three terms of $L_G$ were:

$$L_{Adv}\left(D; G\right) = E_{(x,s)} \left[ \left(D\left(x\right) - 1\right)^2 + \left(D\left(x_{SAE}\right)\right)^2 \right]$$

$$L_{Mel}\left(G\right) = E_{(x,s)} \left[ \left\| \varphi\left(x\right) - \varphi\left(x_{SAE}\right) \right\|_1 \right]$$

$$L_{FM}\left(G, D\right) = E_{(x,s)} \left[ \sum_{i=1}^{T} \frac{1}{N_i} \left\| D^i\left(x\right) - D^i\left(x_{SAE}\right) \right\|_1 \right]$$

Where $\varphi$ represented the mel-spectrogram operator, $T$ was the number of layers in the discriminator, $D^i$ and $N_i$ were the features and the number of features in the $i$th layer of the discriminator, $x$ and $x_{SAE}$ represented the original speech and the reconstructed speech using the pre-trained speech Autoencoder:

$$x_{SAE} = G\left(E\left(x\right)\right)$$

The parameters in Wav2Vec2.0 were frozen within this training phase. The parameters in HiFi-GAN were optimized using the Adam optimizer with a fixed learning rate of $10^{-5}$, $\beta_1 = 0.9$, $\beta_2 = 0.999$. We trained this Autoencoder in LibriSpeech, a 960 hr English speech corpus with a sampling rate of 16 kHz, which is entirely separate from the TIMIT corpus used for our ECoG experiments. We spent 12 days in parallel training on 6 Nvidia GeForce RTX3090 GPUs. The maximum training epoch was 2000. The optimization did not stop until the validation loss no longer decreased.

## Acoustic feature adaptor training and ablation test

The acoustic feature adaptor was trained to acquire the acoustically fidelity re-synthesized natural speech. The acoustic speech re-synthesizer received $z$-scored high gamma ECoG snippets with dimensions $N \times T$, where $N$ was the number of selected speech-responsive electrodes, and $T$ was the ECoG recordings (50 Hz) down-sampled to the same sample rate as the speech features extracted from pre-trained Wav2Vec2.0.

A three-layer bidirectional LSTM adaptor $f$ (see *Figure 1B*) aligned recorded ECoG signals to speech features. The output of the adaptor was $768 \times T$, as the input of the HiFi-GAN generator pre-trained in phase 1. The ultimate output of the HiFi-GAN was a speech waveform with a length of $1 \times \left(320T + 80\right)$ in a 16 K sample rate.

We froze all the parameters of the pre-trained speech autoencoder while training the adaptor. The loss function used in this phase consisted of two parts: the mel-spectrogram loss $L_{Mel}$ and the Laplacian loss $L_{Lap}$:

$$L = L_{Mel} + \lambda L_{Lap}$$

where the regularization coefficient $\lambda$ was set as 0.1 using the ablation test.

$L_{Mel}$ was used to enhance the acoustic fidelity and intelligibility of the ultimate re-synthesized speech from recorded neural activity:

$$L_{Mel} = \left\| \Phi\left(x\right) - \Phi\left(x_a\right) \right\|_1$$

where $x$ and $x_a$, respectively represented the original and acoustic fidelity speech, and $\Phi$ represented the mel-spectrogram operator.

In order to improve the phonetic intelligibility, a Laplace operator, as a convolution kernel was applied to the mel-spectrogram, which was a simplified representation of the convolution on the spectrogram, while convolution on the spectrogram was proven to be effective on speech denoising and enhancement (*Xuhong et al., 2021*):

$$L_{Lap} = \left\| \nabla\left(\Phi\left(x\right)\right) - \nabla\left(\Phi\left(x_a\right)\right) \right\|_1$$

The parameters in the HiFi-GAN generator were frozen within this training phase. The parameters in the adaptor were trained in Stochastic Gradient Descent (SGD) optimizer with a fixed learning rate of $3 \times 10^{-3}$ and a momentum of 0.5. A 10% dropout layer was added in this stage to avoid overfitting. The optimization did not stop until the validation loss no longer decreased.

For each participant, we trained a personalized feature adaptor. Each individual feature adapter underwent training for 500 epochs on one NVidia GeForce RTX3090 GPU, which required 12 hr to complete. The dataset was divided into training, validation, and test sets, comprising 70%, 20%, and 10% of the total trials, respectively. The choice of the LSTM architecture over alternatives was informed by an ablation study presented in *Supplementary file 1*.

## Linguistic feature adaptor training and ablation test

The linguistic feature adaptor was designed to decode word token sequences from ECoG recordings. The model received z-scored high gamma ECoG snippets with dimensions ($N \times T$), where $N$ was the number of selected speech-responsive electrodes, and $T$ was the ECoG recordings (50 Hz) down-sampled to the same sample rate as the acoustic feature training stage.

An attention-based Seq2Seq Transformer architecture (see *Figure 1B*) was employed to adapt ECoG signals to linguistic features. The linguistic adaptor consisted of three encoder layers and three decoder layers, with a hidden dimension of 256 and eight attention heads. The input ECoG signals were first projected into a latent space using a linear layer, followed by positional encoding to capture temporal dependencies. The decoder generated word token sequences autoregressively, starting with a start-of-sequence (SOS) token. The output dimension matched the linguistic feature space (1024-dimensional).

The loss function combined the token-level KL-divergence and sequence length prediction loss with L2 regularization:

$$L = L_{token} + \lambda_1 L_{length} + \lambda_2 \left\| p \right\|_2^2$$

where $L_{token}$ measured the KL-divergence between predicted and target token distributions:

$$L_{token} = \text{KL}\left(t_{pred}, t_{target}\right)$$

and $L_{length}$ was a Huber loss between predicted and actual sequence lengths:

$$L_{length} = \text{Huber}\left(l_{pred}, l_{target}\right)$$

The regularization coefficient $\lambda_1$ was set to 1 based on ablation tests, and L2 regularization (weight decay = 0.01) was applied to mitigate overfitting. The model was trained using the Adam optimizer with a fixed learning rate of $10^{-4}$ for 500 epochs. The dataset was split into training (70%), validation (20%), and test (10%) sets, consistent with the acoustic feature adaptor. For each participant, a personalized linguistic adaptor was trained on one NVidia GeForce RTX3090 GPU, requiring approximately

6 hr to complete. The Transformer architecture was selected based on its superior performance in word token prediction compared to recurrent networks, as detailed in *Supplementary file 2*.

### Automatic phoneme recognition of re-synthesized speech sentences

In order to quantitatively evaluate the synthesized speech and recognize the phonemes, we fine-tuned a phoneme recognizer using Wav2Vec2.0, a logistic regression phoneme classifier, and a phoneme decoder. Wav2Vec2.0 was used for feature extraction, the classifier outputs a probability vector from Wav2Vec2.0 features, and the phoneme decoder decoded a phoneme sequence from the probability vectors to finally get the phoneme sequence.

The models are trained and evaluated on the training set and test set of TIMIT and tested on the re-synthesized ECoG recordings. These parameters were trained using the AdamW optimizer with a fixed learning rate of $10^{-4}$, $\beta_1 = 0.9$, $\beta_2 = 0.999$, and $\varepsilon = 10^{-8}$ in the CTC loss function. The optimization did not stop until the validation loss no longer decreased.

### Phoneme error rate (PER) and word error rate (WER)

We implemented a two-stage transcription pipeline to evaluate linguistic intelligibility. First, word sequences were transcribed using OpenAI's Whisper model (*Radfo et al., 2023*) (base version) with default parameters, which provided both word-level alignments and confidence scores. The reference and decoded word sequences were then aligned using edit distance to compute WER. The error rates were calculated by applying the same edit distance algorithm to the forced-aligned phoneme sequences. This combined approach ensured consistent phoneme representations between natural and synthesized speech during comparison:

$$\text{error rate} = \frac{\text{Edit}\left(original, transcribed\right)}{N}$$

Where $\text{Edit}\left(original, transcribed\right)$ denotes the edit distance between original and transcribed phoneme sequences; $N$ is the number of phonemes in the sentence.

### Alignment of the ultimate re-synthesized speech to calculate the mel-spectrogram $R^2$

To quantitatively evaluate the acoustic fidelity of the re-synthesized speech using mel-spectrogram $R^2$, we performed temporal alignment between the reconstructed and ground truth mel-spectrograms. First, word-level timestamps were extracted from the transcribed speech using OpenAI's Whisper model, which provided precise boundaries for voiced segments. The mel-spectrogram of the re-synthesized speech was then segmented into these intervals, excluding silent or non-speech regions. Each voiced segment was resampled to match the temporal resolution of the corresponding ground truth spectrogram using linear interpolation, ensuring frame-by-frame comparability. The squared Pearson correlation coefficient ($R^2$) was computed over the aligned spectrograms, measuring the proportion of variance in the ground truth spectrogram explained by the reconstructed features. This approach accounted for natural variations in speech rate while rigorously quantifying spectral fidelity, with higher $R^2$ values (closer to 1) indicating superior preservation of time-frequency structures critical for speech intelligibility and naturalness.

### Voice cloning for the ultimate re-synthesized natural speech

To achieve personalized speech output, we implemented a voice cloning pipeline that preserves the subject's unique vocal features. The system takes two key inputs: (1) a clean voice sample from the denoised linguistic speech reference, and (2) the corresponding word sequence transcribed by Whisper. This approach serves dual purposes: Whisper first transcribes the original speech to provide accurate text prompts, then later evaluates the intelligibility of the ultimate re-synthesized speech by transcribing it again for word error rate calculation.

The cloning process begins with extracting vocal fingerprints from the short reference sample, capturing distinctive timbral qualities like pitch contour and formant structure. These vocal features are then mapped to the target word sequences, generating synthetic speech that maintains the subject's natural voice patterns while clearly articulating the desired content.

We specifically employed the '3 s rapid cloning' mode to optimize for both efficiency and voice quality. This ultra-fast processing allows for quick adaptation to individual speakers while still producing highly natural-sounding results. The synthesized output was rigorously evaluated through both objective metrics (Whisper-based transcription accuracy) and subjective listening tests to ensure faithful reproduction of the original voice features.

## Fine-tuning CosyVoice2.0 on the TIMIT corpus

To enhance the fidelity and intelligibility of re-synthesized speech while utilizing the generative capacity of large speech models for artifact mitigation, CosyVoice2.0 was fine-tuned on the sentences from the TIMIT corpus used in the training set of neural-driven dual-pathway model training. The architecture comprises two core components: (i) a 'Flow' module converting tokens to acoustic features via Text Embedding, Speaker Embedding (linear layer), Encoder, and causal conditional CFM Decoder; and (ii) a 'Hift' vocoder synthesizing waveforms from Flow outputs using an F0 extractor, Up-sampler, and nine residual blocks. Selective parameter optimization was applied to: (i) the first and final layers of the 12-layer Flow decoder, and (ii) the Speaker Embedding linear layer. The model mapped speech autoencoder outputs and punctuation-free plain text to ground-truth waveforms, optimized with a compound loss consisting of mel-spectrogram KL-divergence and MFCC loss:

$$L = \text{KL}\left(\Phi\left(x_{gt}\right),\, \Phi\left(x_{output}\right)\right) + \left\|\text{MFCC}\left(x_{gt}\right) - \text{MFCC}\left(x_{output}\right)\right\|_1$$

These models were trained using the AdamW optimizer with a fixed learning rate of $10^{-5}$, and set $\beta_1 = 0.9$, $\beta_2 = 0.999$, and $\varepsilon = 10^{-8}$. $x$ represented the speech waveform, and $\Phi$ represented the mel-spectrogram operator.

## Ablation study on training data volume

To assess the impact of training data quantity on decoding performance, we conducted an additional ablation experiment. For each participant, we created subsets of the full training set corresponding to 25%, 50%, and 75% of the original data by random sampling while preserving the temporal continuity of speech segments. Personalized acoustic and linguistic adaptors were then independently trained from scratch on each subset, following the identical architecture and optimization procedures described above. All other components of the pipeline, including the frozen pre-trained generators (HiFi-GAN, Parler-TTS) and the CosyVoice 2.0 voice cloner, remained unchanged. Performance metrics (mel-spectrogram R², WER, PER) were evaluated on the same held-out test set for all data conditions. The results (*Figure 3—figure supplement 2*) demonstrate that when more than 50% of the training data is utilized, performance degrades gracefully rather than catastrophically, which is a promising indicator for clinical applications with limited data collection time.

## Acknowledgements

This work is supported by the National Natural Science Foundation of China (32371154, YL), the National Science and Technology Major Project of China (2025ZD0217000, YL), Shanghai Rising-Star Program (24QA2705500, YL), and the Lin Gang Laboratory (LG-GG-202402–06 and LGL-1987–18, YL). The computations in this research are supported by the HPC Platform of ShanghaiTech University.

## Additional information

### Competing interests

Edward F Chang: is the co-founder of Echo Neurotechnologies, LLC. The other authors declare that no competing interests exist.

### Funding

| Funder | Grant reference number | Author |
|---|---|---|
| National Natural Science Foundation of China | 32371154 | Yuanning Li |

| Funder | Grant reference number | Author |
|---|---|---|
| National Science and Technology Major Project | 2025ZD0217000 | Yuanning Li |
| Science and Technology Commission of Shanghai Municipality | 24QA2705500 | Yuanning Li |
| Lin Gang Laboratory | LG-GG-202402-06 | Yuanning Li |
| Lin Gang Laboratory | LGL-1987-18 | Yuanning Li |

The funders had no role in study design, data collection and interpretation, or the decision to submit the work for publication.

### Author contributions

Jiawei Li, Software, Formal analysis, Investigation, Methodology, Writing - original draft, Writing - review and editing; Chunxu Guo, Software, Methodology; Chao Zhang, Methodology; Edward F Chang, Resources; Yuanning Li, Conceptualization, Resources, Software, Formal analysis, Supervision, Investigation, Methodology, Writing - original draft, Project administration, Writing - review and editing

### Author ORCIDs

Edward F Chang ⬤ https://orcid.org/0000-0003-2480-4700
Yuanning Li ⬤ https://orcid.org/0000-0003-3277-0600

### Ethics

All protocols in the current study were approved by the Institutional Review Board (IRB) of the University of California, San Francisco. Participants were thoroughly informed about the experiment and research and gave written consent. They volunteered for participation, ensuring that their involvement did not influence their clinical care. Additionally, verbal consent was obtained from participants at the beginning of each experimental session.

Reviewer #1 (Public review): https://doi.org/10.7554/eLife.109400.3.sa1
Reviewer #2 (Public review): https://doi.org/10.7554/eLife.109400.3.sa2
Author response https://doi.org/10.7554/eLife.109400.3.sa3

## Additional files

### Supplementary files

Supplementary file 1. Ablation study on adaptor architecture for the acoustic pathway. Performance is evaluated by the mel-spectrogram $R^2$ (mean ± s.e.m. across participants), measuring the fidelity of reconstructed acoustic features. The bidirectional long-short term memory (LSTM) adaptor consistently outperformed the Transformer-based adaptor across different layer depths. The optimal performance was achieved with a 3-layer LSTM, which was selected for the final model.

Supplementary file 2. Ablation study on adaptor architecture for the linguistic pathway. Performance is evaluated by word error rate (WER) and phoneme error rate (PER) (mean ± s.e.m. across participants), measuring the intelligibility of reconstructed speech. The Transformer-based adaptor achieved lower error rates than the long-short term memory (LSTM)-based adaptor across nearly all layer configurations. The optimal performance was achieved with a 3-layer Transformer, which was selected for the final model.

Supplementary file 3. Comparative overview of recent studies in neural-driven speech decoding and re-synthesis. The table summarizes representative work, highlighting the neural recording modality, approximate amount of data used for decoder training per subject, the primary experimental task (perception or production), and reported performance metrics. Studies are ordered chronologically. Performance metrics include: Mean Opinion Score (MOS, scale 1–5), Extended Short-Time Objective Intelligibility (ESTOI, scale 0–1), Word Error Rate (WER, %), Phoneme Error Rate (PER, %), and mel-spectrogram correlation ($R^2$, scale 0–1). Note that direct numerical comparisons should be made with caution due to differences in neural signals, tasks, stimuli, and evaluation methodologies across studies. Our study (highlighted in bold) achieves a competitive balance between data efficiency

(~20 min) and performance across multiple metrics (WER, PER, MOS, $R^2$).

MDAR checklist

Source data 1. The statistical source data for all figures and supplementary figures.

### Data availability

The data that support the findings of this study are available on request from the lead contact. The data are not publicly available because they could compromise research participant privacy and consent. All original code and preprocessed anonymized data to replicate the main findings of this study can be found at https://github.com/CCTN-BCI/Neural2Speech2, copy archived at *CCTN-BCI, 2026*. Any additional information required to reanalyze the data reported in this paper is available from the lead contact upon request.

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
