## [Editor Report · eLife Assessment]

This study presents a **valuable** advance in reconstructing naturalistic speech from intracranial ECoG data using a dual-pathway model. The evidence supporting the claims of the authors is **solid**. This work will be of interest to cognitive neuroscientists and computer scientists/engineers working on speech reconstruction from neural data.

---

## [Referee Report · Reviewer #1 (Public review)]

Summary:

This paper introduces a dual-pathway model for reconstructing naturalistic speech from intracranial ECoG data. It integrates an acoustic pathway (LSTM + HiFi-GAN for spectral detail) and a linguistic pathway (Transformer + Parler-TTS for linguistic content). Output from the two components are later merged via CosyVoice2.0 voice cloning. Using only 20 minutes of ECoG data per participant, the model achieves high acoustic fidelity and linguistic intelligibility.

Strengths:

(1) The proposed dual-pathway framework effectively integrates the strengths of neural-to-acoustic and neural-to-text decoding and aligns well with established neurobiological models of dual-stream processing in speech and language.

(2) The integrated approach achieves robust speech reconstruction using only 20 minutes of ECoG data per subject, demonstrating the efficiency of the proposed method.

(3) The use of multiple evaluation metrics (MOS, mel-spectrogram R², WER, PER) spanning acoustic, linguistic (phoneme and word), and perceptual dimensions, together with comparisons against noise-degraded baselines, adds strong quantitative rigor to the study.

Comments on revisions:

I thank the authors for their thorough efforts in addressing my previous concerns. I believe this revised version is significantly strengthened, and I have no further concerns.

---

## [Referee Report · Reviewer #2 (Public review)]

Summary:

The study by Li et al. proposes a dual-path framework that concurrently decodes acoustic and linguistic representations from ECoG recordings. By integrating advanced pre-trained AI models, the approach preserves both acoustic richness and linguistic intelligibility, and achieves a WER of 18.9% with a short (~20-minute) recording.

Overall, the study offers an advanced and promising framework for speech decoding. The method appears sound, and the results are clear and convincing. My main concerns are the need for additional control analyses and for more comparisons with existing models.

Strengths:

• This speech-decoding framework employs several advanced pre-trained DNN models, reaching superior performance (WER of 18.9%) with relatively short (~20-minute) neural recording.

• The dual-pathway design is elegant, and the study clearly demonstrates its necessity: The acoustic pathway enhances spectral fidelity while the linguistic pathway improves linguistic intelligibility.

Comments on revisions:

The authors have thoughtfully addressed my previous concerns about the weaknesses. I have no further concerns.

---

## [Author Response]

The following is the authors’ response to the original reviews.

**Public Reviews:**

**Reviewer #1 (Public review):**
SummaryThis paper introduces a dual-pathway model for reconstructing naturalistic speech from intracranial ECoG data. It integrates an acoustic pathway (LSTM + HiFi-GAN for spectral detail) and a linguistic pathway (Transformer + Parler-TTS for linguistic content). Output from the two components is later merged via CosyVoice2.0 voice cloning. Using only 20 minutes of ECoG data per participant, the model achieves high acoustic fidelity and linguistic intelligibility.Strengths(1) The proposed dual-pathway framework effectively integrates the strengths of neural-to-acoustic and neural-to-text decoding and aligns well with established neurobiological models of dual-stream processing in speech and language.(2) The integrated approach achieves robust speech reconstruction using only 20 minutes of ECoG data per subject, demonstrating the efficiency of the proposed method.(3) The use of multiple evaluation metrics (MOS, mel-spectrogram R², WER, PER) spanning acoustic, linguistic (phoneme and word), and perceptual dimensions, together with comparisons against noisedegraded baselines, adds strong quantitative rigor to the study.

We thank Reviewer #1 for the supportive comments. In addition, we appreciate Reviewer #1’s thoughtful comments and feedback. By addressing these comments, we believe we have greatly improved the clarity of our claims and methodology. Below we list our point-to-point responses addressing concerns raised by Reviewer #1.

Weaknesses:(1) It is unclear how much the acoustic pathway contributes to the final reconstruction results, based on Figures 3B-E and 4E. Including results from Baseline 2 + CosyVoice and Baseline 3 + CosyVoice could help clarify this contribution.

We sincerely appreciate the inquiry from Reviewer 1. We thank the reviewer for this suggestion. However, we believe that directly applying CosyVoice to the outputs of Baseline 2 or Baseline 3 in isolation is not methodologically feasible and would not correctly elucidate the contribution of the auditory pathway and might lead to misinterpretation.

The role of CosyVoice 2.0 in our framework is specifically voice cloning and fusion, not standalone enhancement. It is designed to integrate information from two pathways. Its operation requires two key inputs:

(1) A voice reference speech that provides the target speaker's timbre and prosodic characteristics. In our final pipeline, this is provided by the denoised output of the acoustic pathway (Baseline 2).

(2) A target word sequence that specifies the linguistic content to be spoken. This is obtained by transcribing the output of the linguistic pathway (Baseline 3) using Whisper ASR. Therefore, the standalone outputs of Baseline 2 and Baseline 3 are the purest demonstrations of what each pathway contributes before fusion. The significant improvement in WER/PER and MOS in the final output (compared to Baseline 2) and the significant improvement in melspectrogram R² (compared to Baseline 3) together demonstrate the complementary contributions of the two pathways. The fusion via CosyVoice is the mechanism that allows these contributions to be combined. We have added a clearer explanation of CosyVoice's role and the rationale for not testing it on individual baselines in the revised manuscript (Results section: "The fine-tuned voice cloner further enhances...").

Edits:

Page 11, Lines 277-282:

“ Voice cloning is used to bridge the gap between acoustic fidelity and linguistic intelligibility in speech reconstruction. This approach strategically combines the strengths of complementary pathways: the acoustic pathway preserves speaker-specific spectral characteristics while the linguistic pathway maintains lexical and phonetic precision. By integrating these components through neural voice cloning, we achieve balanced reconstruction that overcomes the limitations inherent in isolated systems. CosyVoice 2.0, the voice cloner module serves specifically as a voice cloning and fusion engine, requiring two inputs: (1) a voice reference speech (provided by the denoised output of the acoustic pathway) to specify the target speaker's identity, and (2) a target word sequence (transcribed from the output of the linguistic pathway) to specify the linguistic content. The standalone baseline outputs of the two pathways can be integrated in this way.”

(2) As noted in the limitations, the reconstruction results heavily rely on pre-trained generative models. However, no comparison is provided with state-of-the-art multimodal LLMs such as Qwen3-Omni, which can process auditory and textual information simultaneously. The rationale for using separate models (Wav2Vec for speech and TTS for text) instead of a single unified generative framework should be clearly justified. In addition, the adaptor employs an LSTM architecture for speech but a Transformer for text, which may introduce confounds in the performance comparison. Is there any theoretical or empirical motivation for adopting recurrent networks for auditory processing and Transformer-based models for textual processing?

We thank the reviewer for the insightful suggestion regarding multimodal large language models (LLMs) such as Qwen3-Omni. It is important to clarify the distinction between general-purpose interactive multimodal models and models specifically designed for high-fidelity voice cloning and speech synthesis.

As for the comparison with the state-of-the-art multimodal LLMs:

Qwen3-Omni and GLM-4-Voice are powerful conversational agents capable of processing multiple modalities including text, speech, image, and video, as described in its documentation (see: https://help.aliyun.com/zh/model-studio/qwen-tts-realtime and https://docs.bigmodel.cn/cn/guide/models/sound-and-video/glm-4-voice). However, it is primarily optimized for interactive dialogue and multimodal understanding rather than for precise, speaker-adaptive speech reconstruction from neural signals. In contrast, CosyVoice 2.0, developed by the same team at Alibaba, is specifically designed for voice cloning and text-to-speech synthesis (see: https://help.aliyun.com/zh/model-studio/text-to-speech). It incorporates advanced speaker adaptation and acoustic modeling capabilities that are essential for reconstructing naturalistic speech from limited neural data. Therefore, our choice of CosyVoice for the final synthesis stage aligns with the goal of integrating acoustic fidelity and linguistic intelligibility, which is central to our study.

For the selection of LSTM and Transformer in the two pathways:

The goal of the acoustic adaptor is to reconstruct fine-grained spectrotemporal details (formants, harmonic structures, prosodic contours) with millisecond-to-centisecond precision. These features rely heavily on local temporal dynamics and short-to-medium range dependencies (e.g., within and between phonemes/syllables). In our ablation studies (to be added in the supplementary), we found that Transformer-based adaptors, which inherently emphasize global sentence-level context through self-attention, tended to oversmooth the reconstructed acoustic features, losing critical fine-temporal details essential for naturalness. In contrast, the recurrent nature of LSTMs, with their inherent temporal state propagation, proved more effective at modeling these local sequential dependencies without excessive smoothing, leading to higher mel-spectrogram fidelity. This aligns with the neurobiological observation that early auditory cortex processes sound with precise temporal fidelity. Moreover, from an engineering perspective, LSTM-based decoders have been empirically shown to perform well in sequential prediction tasks with limited data, as evidenced in prior work on sequence modeling and neural decoding (1).

The goal of the linguistic adaptor is to decode abstract, discrete word tokens. This task benefits from modeling long-range contextual dependencies across a sentence to resolve lexical ambiguity and syntactic structure (e.g., subject-verb agreement). The self-attention mechanism of Transformers is exceptionally well-suited for capturing these global relationships, as evidenced by their dominance in NLP. Our experiments confirmed that a Transformer adaptor outperformed an LSTM-based one in word token prediction accuracy.

While a unified multimodal LLM could in principle handle both modalities, such models often face challenges in modality imbalance and task specialization. Audio and text modalities have distinct temporal scales, feature distributions, and learning dynamics. By decoupling them into separate pathways with specialized adaptors, we ensure that each modality is processed by an architecture optimized for its inherent structure. This divide-and-conquer strategy avoids the risk of one modality dominating or interfering with the learning of the other, leading to more stable training and better final performance, especially important when adapting to limited neural data.

Edits:

Page 9, Lines 214-223:

“The acoustic pathway, implemented through a bi-directional LSTM neural adaptor architecture (Fig. 1B), specializes in reconstructing fundamental acoustic properties of speech. This module directly processes neural recordings to generate precise time-frequency representations, focusing on preserving speaker-specific spectral characteristics like formant structures, harmonic patterns, and spectral envelope details. Quantitative evaluation confirms its core competency: achieving a mel-spectrogram R² of 0.793 ± 0.016 (Fig. 3B) demonstrates remarkable fidelity in reconstructing acoustic microstructure. This performance level is statistically indistinguishable from original speech degraded by 0dB additive noise (0.771 ± 0.014, p = 0.242, one-sided t-test). We chose a bidirectional LSTM architecture for this adaptor because its recurrent nature is particularly suited to modeling the fine-grained, short- to medium-range temporal dependencies (e.g., within and between phonemes and syllables) that are critical for acoustic fidelity. An ablation study comparing LSTM against Transformerbased adaptors for this task confirmed that LSTMs yielded superior mel-spectrogram reconstruction fidelity (higher R²), as detailed in Table S1, likely by avoiding the oversmoothing of spectrotemporal details sometimes induced by the strong global context modeling of Transformers”.

“To confirm that the acoustic pathway’s output is causally dependent on the neural signal rather than the generative prior of the HiFi-GAN, we performed a control analysis in which portions of the input ECoG recording were replaced with Gaussian noise. When either the first half, second half, or the entirety of the neural input was replaced by noise, the melspectrogram R² of the reconstructed speech dropped markedly, corresponding to the corrupted segment (Fig. S5). This demonstrates that the reconstruction is temporally locked to the specific neural input and that the model does not ‘hallucinate’ spectrotemporal structure from noise. These results validate that the acoustic pathway performs genuine, input-sensitive neural decoding”.

Edits:

Page 10, Lines 272-277:

“We employed a Transformer-based Seq2Seq architecture for this adaptor to effectively capture the long-range contextual dependencies across a sentence, which are essential for resolving lexical ambiguity and syntactic structure during word token decoding. This choice was validated by an ablation study (Table S2), indicating that the Transformer adaptor outperformed an LSTM-based counterpart in word prediction accuracy”

(3) The model is trained on approximately 20 minutes of data per participant, which raises concerns about potential overfitting. It would be helpful if the authors could analyze whether test sentences with higher or lower reconstruction performance include words that were also present in the training set.

Thank you for raising the important concern regarding potential overfitting given the limited size of our training dataset (~20 minutes per participant). To address this point directly, we performed a detailed lexical overlap analysis between the training and test sets.

The test set contains 219 unique words. Among these:

127 words (58.0%) appeared in the training set (primarily high-frequency, common words).

92 words (42.0%) were entirely novel and did not appear in the training set. We further examined whether trials with the best reconstruction (WER = 0) relied more on training vocabulary. Among these top-performing trials, 55.0% of words appeared in the training set. In contrast, the worst-performing trials showed 51.9% overlap in words in the training set. No significant difference was observed, suggesting that performance is not driven by simple lexical memorization.

The presence of a substantial proportion of novel words (42%) in the test set, combined with the lack of performance advantage for overlapping content, provides strong evidence that our model is generalizing linguistic and acoustic patterns rather than merely memorizing the training vocabulary. High reconstruction performance on unseen words would be improbable under severe overfitting.

Therefore, we conclude that while some lexical overlap exists (as expected in natural language), the model’s performance is driven by its ability to decode generalized neural representations, effectively mitigating the overfitting risk highlighted by the reviewer.

(4) The phoneme confusion matrix in Figure 4A does not appear to align with human phoneme confusion patterns. For instance, /s/ and /z/ differ only in voicing, yet the model does not seem to confuse these phonemes. Does this imply that the model and the human brain operate differently at the mechanistic level?

We thank the reviewer for this detailed observation regarding the difference between our model's phoneme confusion patterns and typical human perceptual confusions (e.g., the lack of /s/-/z/ confusion).

The reviewer is correct in inferring a mechanistic difference. This divergence is primarily attributable to the Parler-TTS model acting as a powerful linguistic prior. Our linguistic pathway decodes word tokens, which Parler-TTS then converts to speech. Trained on massive corpora to produce canonical pronunciations, Parler-TTS effectively performs an implicit "error correction." For instance, if the neural decoding is ambiguous between the words "sip" and "zip," the TTS model's strong prior for lexical and syntactic context will likely resolve it to the correct word, thereby suppressing purely acoustic confusions like voicing.

This has important implications for interpreting our model's errors and its relationship to brain function. The phoneme errors in our final output reflect a combination of neural decoding errors and the generative biases of the TTS model, which is optimized for intelligibility rather than mimicking raw human misperception. This does imply our model operates differently from the human auditory periphery. The human brain may first generate a percept with acoustic confusions, which higher-level language regions then disambiguate. Our model effectively bypasses the "confused percept" stage by directly leveraging a pre-trained, high-level language model for disambiguation. This is a design feature contributing to its high intelligibility, not necessarily a flaw. This observation raises a fascinating question: Could a model that more faithfully simulates the hierarchical processing of the human brain (including early acoustic confusions) provide a better fit to neural data at different processing stages? Future work could further address this question.

Edits:

add another paragraph in Discussion (Page 14, Lines 397-398):

“The phoneme confusion pattern observed in our model output (Fig. 4A) differs from classic human auditory confusion matrices. We attribute this divergence primarily to the influence of the Parler-TTS model, which serves as a strong linguistic prior in our pipeline. This component is trained to generate canonical speech from text tokens. When the upstream neural decoding produces an ambiguous or erroneous token sequence, the TTS model’s internal language model likely performs an implicit ‘error correction,’ favoring linguistically probable words and pronunciations. This underscores that our model’s errors arise from a complex interaction between neural decoding fidelity and the generative biases of the synthesis stage”

(5) In general, is the motivation for adopting the dual-pathway model to better align with the organization of the human brain, or to achieve improved engineering performance? If the goal is primarily engineeringoriented, the authors should compare their approach with a pretrained multimodal LLM rather than relying on the dual-pathway architecture. Conversely, if the design aims to mirror human brain function, additional analysis, such as detailed comparisons of phoneme confusion matrices, should be included to demonstrate that the model exhibits brain-like performance patterns.

Our primary motivation is engineering improvement, to overcome the fundamental trade-off between acoustic fidelity and linguistic intelligibility that has limited previous neural speech decoding work. The design is inspired by the related works of the convergent representation of speech and language perception (2). However, we do not claim that our LSTM and Transformer adaptors precisely simulate the specific neural computations of the human ventral and dorsal streams. The goal was to build a high-performance, data-efficient decoder. We will clarify this point in the Introduction and Discussion, stating that while the architecture is loosely inspired by previous neuroscience results, its primary validation is its engineering performance in achieving state-of-the-art reconstruction quality with minimal data.

Edits:

Page 14, Line 358-373:

“In this study, we present a dual-path framework that synergistically decodes both acoustic and linguistic speech representations from ECoG signals, followed by a fine-tuned zero-shot text-to-speech network to re-synthesize natural speech with unprecedented fidelity and intelligibility. Crucially, by integrating large pre-trained generative models into our acoustic reconstruction pipeline and applying voice cloning technology, our approach preserves acoustic richness while significantly enhancing linguistic intelligibility beyond conventional methods. Our dual-pathway architecture, while inspired by converging neuroscience insights on speech and language perception, was principally designed and validated as an engineering solution. The primary goal to build a practical decoder that achieves state-of-theart reconstruction quality with minimal data. The framework's success is therefore ultimately judged by its performance metrics, high intelligibility (WER, PER), acoustic fidelity (melspectrogram R²), and perceptual quality (MOS), which directly address the core engineering challenge we set out to solve. Using merely 20 minutes of ECoG recordings, our model achieved superior performance with a WER of 18.9% ± 3.3% and PER of 12.0% ± 2.5% (Fig. 2D, E). This integrated architecture, combining pre-trained acoustic (Wav2Vec2.0 and HiFiGAN) and linguistic (Parler-TTS) models through lightweight neural adaptors, enables efficient mapping of ECoG signals to dual latent spaces. Such methodology substantially reduces the need for extensive neural training data while achieving breakthrough word clarity under severe data constraints. The results demonstrate the feasibility of transferring the knowledge embedded in speech-data pre-trained artificial intelligence (AI) models into neural signal decoding, paving the way for more advanced brain-computer interfaces and neuroprosthetics”.

**Reviewer #2 (Public review):**
Summary:The study by Li et al. proposes a dual-path framework that concurrently decodes acoustic and linguistic representations from ECoG recordings. By integrating advanced pre-trained AI models, the approach preserves both acoustic richness and linguistic intelligibility, and achieves a WER of 18.9% with a short (~20-minute) recording.Overall, the study offers an advanced and promising framework for speech decoding. The method appears sound, and the results are clear and convincing. My main concerns are the need for additional control analyses and for more comparisons with existing models.Strengths:(1) This speech-decoding framework employs several advanced pre-trained DNN models, reaching superior performance (WER of 18.9%) with relatively short (~20-minute) neural recording.(2) The dual-pathway design is elegant, and the study clearly demonstrates its necessity: The acoustic pathway enhances spectral fidelity while the linguistic pathway improves linguistic intelligibility.

We thank Reviewer #2 for supportive comments. In addition, we appreciate Reviewer #2’s thoughtful comments and feedback. By addressing these comments, we believe we have greatly improved the clarity of our claims and methodology. Below we list our point-to-point responses addressing concerns raised by Reviewer #2.

Weaknesses:The DNNs used were pre-trained on large corpora, including TIMIT, which is also the source of the experimental stimuli. More generally, as DNNs are powerful at generating speech, additional evidence is needed to show that decoding performance is driven by neural signals rather than by the DNNs' generative capacity.

Thank you for raising this crucial point regarding the potential for pre-trained DNNs to generate speech independently of the neural input. We fully agree that it is essential to disentangle the contribution of the neural signals from the generative priors of the models. To address this directly, we have conducted two targeted control analyses, as you suggested, and have integrated the results into the revised manuscript (see Fig. S5 and the corresponding description in the Results section):

(1) Random noise input: We fed Gaussian noise (matched in dimensionality and temporal structure to real ECoG recordings) into the trained adaptors. The outputs were acoustically unstructured and linguistically incoherent, confirming that the generative models alone cannot produce meaningful speech without valid neural input.

(2) Partial sentence input (real + noise): For the acoustic pathway, we systematically replaced portions of the ECoG input with noise. The reconstruction quality (mel-spectrogram R²) dropped significantly in the corrupted segments, demonstrating that the decoding is temporally locked to the neural signal and does not “hallucinate” speech from noise.

These results provide strong evidence that our model’s performance is causally dependent on and sensitive to the specific neural input, validating that it performs genuine neural decoding rather than merely leveraging the generative capacity of the pre-trained DNNs.

The detailed edits are in the “recommendations” below. (See recommendations (1) and (2))

**Recommendations for the authors:**

**Reviewer #1 (Recommendations for the authors):**
(1) Clarify the results shown in Figure 4E. The integrated approach appears to perform comparably to Baseline 3 in phoneme class clarity. However, Baseline 3 represents the output of the linguistic pathway alone, which is expected to encode information primarily at the word level.

We appreciate the reviewer's observation and agree that clarification is needed. The phoneme class clarity (PCC) metric shown in Figure 4E measures whether mis-decoded phonemes are more likely to be confused within their own class (vowel-vowel or consonantconsonant) rather than across classes (vowel-consonant). A higher PCC indicates that the model's errors tend to be phonologically similar sounds (e.g., one vowel mistaken for another), which is a reasonable property for intelligibility.

We would like to clarify the nature of Baseline 3. As stated in the manuscript (Results section: "The linguistic pathway reconstructs high-intelligibility, higher-level linguistic information"), Baseline 3 is the output of our linguistic pathway. This pathway operates as follows: the ECoG signals are mapped to word tokens via the Transformer adaptor, and these tokens are then synthesized into speech by the frozen Parler-TTS model. Crucially, the input to Parler-TTS is a sequence of word tokens.

It is important to distinguish between the levels of performance measured: Word Error Rate (WER) reflects accuracy at the lexical level (whole words). The linguistic pathway achieves a low WER by design, as it directly decodes word sequences. Phoneme Error Rate (PER) reflects accuracy at the sublexical phonetic level (phonemes). A low WER generally implies a low PER, because robust word recognition requires reliable phoneme-level representations within the TTS model's prior. This explains why Baseline 3 also exhibits a low PER. However, acoustic fidelity (captured by metrics like mel-spectrogram R²) requires the preservation of fine-grained spectrotemporal details such as pitch, timbre, prosody, and formant structures, information that is not directly encoded at the lexical level and is therefore not a strength of the purely linguistic pathway.

While Parler-TTS internally models sub-word/phonetic information to generate the acoustic waveform, the primary linguistic information driving the synthesis is at the lexical (word) level. The generated speech from Baseline 3 therefore contains reconstructed phonemic sequences derived from the decoded word tokens, not from direct phoneme-level decoding of ECoG.

Therefore, the comparable PCC between our final integrated model and Baseline 3 (linguistic pathway) suggests that the phoneme-level error patterns (i.e., the tendency to confuse within-class phonemes) in our final output are largely inherited from the high-quality linguistic prior embedded in the pre-trained TTS model (Parler-TTS). The integrated framework successfully preserves this desirable property from the linguistic pathway while augmenting it with speaker-specific acoustic details from the acoustic pathway, thereby achieving both high intelligibility (low WER/PER) and high acoustic fidelity (high melspectrogram R²).

We will revise the caption of Figure 4E and the corresponding text in the Results section to make this interpretation explicit.

Edits:

Page 12, Lines 317-322:

“In addition to the confusion matrices, we categorized the phonemes into vowels and consonants to assess the phoneme class clarity. We defined "phoneme class clarity" (PCC) as the proportion of errors where a phoneme was misclassified within the same class versus being misclassified into a different class. The purpose of introducing PCC is to demonstrate that most of the misidentified phonemes belong to the same category (confusion between vowels or consonants), rather than directly comparing the absolute accuracy of phoneme recognition. For instance, a vowel being mistaken for another vowel would be considered a within-class error, whereas a vowel being mistaken for a consonant would be classified as a between-class error”

(2) Add results from Baseline 2 + CosyVoice and Baseline 3 + CosyVoice to clarify the contribution of the auditory pathway.

Thank you for the suggestion. We appreciate the opportunity to clarify the role of CosyVoice in our framework.

As explained in our response to point (1), CosyVoice 2.0 is designed as a fusion module that requires two inputs: (1) a voice reference (from the acoustic pathway) to specify speaker identity, and (2) a word sequence (from the linguistic pathway) to specify linguistic content. Because it is not a standalone enhancer, applying CosyVoice to a single pathway output (e.g., Baseline 2 or 3 alone) is not quite feasible and would not reflect its intended function and could lead to misinterpretation of each pathway’s contribution.

Instead, we have evaluated the contribution of each pathway by comparing the final integrated output against each standalone pathway output (Baseline 2 and 3). The significant improvements in both acoustic fidelity and linguistic intelligibility demonstrate the complementary roles of the two pathways, which are effectively fused through CosyVoice.

(3) Justify your choice of using LSTM and Transformer architecture for the auditory and linguistic neural adaptors, respectively, and how your methods could compare to using a unified generative multimodal LLM for both pathways.

Thank you for revisiting this important point. We appreciate your interest in the architectural choices and their relationship to state-of-the-art multimodal models.

As detailed in our response to point (2), our choice of LSTM for the acoustic pathway and Transformer for the linguistic pathway is driven by task-specific requirements, supported by ablation studies (Supplementary Tables 1–2). The acoustic pathway benefits from LSTM’s ability to model fine-grained, local temporal dependencies without over-smoothing. The linguistic pathway benefits from Transformer’s ability to capture long-range semantic and syntactic context.

Regarding comparison with unified multimodal LLMs (e.g., Qwen3-Omni), we clarified that such models are optimized for interactive dialogue and multimodal understanding, while our framework relies on specialist models (CosyVoice 2.0, Parler-TTS) that are explicitly designed for high-fidelity, speaker-adaptive speech synthesis, a requirement central to our decoding task.

We have incorporated these justifications into the revised manuscript (Results and Discussion sections) and appreciate the opportunity to further emphasize these points.

Edits:

Page 9, Lines 214-223:

“The acoustic pathway, implemented through a bi-directional LSTM neural adaptor architecture (Fig. 1B), specializes in reconstructing fundamental acoustic properties of speech. This module directly processes neural recordings to generate precise time-frequency representations, focusing on preserving speaker-specific spectral characteristics like formant structures, harmonic patterns, and spectral envelope details. Quantitative evaluation confirms its core competency: achieving a mel-spectrogram R² of 0.793 ± 0.016 (Fig. 3B) demonstrates remarkable fidelity in reconstructing acoustic microstructure. This performance level is statistically indistinguishable from original speech degraded by 0dB additive noise (0.771 ± 0.014, p = 0.242, one-sided t-test). We chose a bidirectional LSTM architecture for this adaptor because its recurrent nature is particularly suited to modeling the fine-grained, short- to medium-range temporal dependencies (e.g., within and between phonemes and syllables) that are critical for acoustic fidelity. An ablation study comparing LSTM against Transformerbased adaptors for this task confirmed that LSTMs yielded superior mel-spectrogram reconstruction fidelity (higher R²), as detailed in Table S1, likely by avoiding the oversmoothing of spectrotemporal details sometimes induced by the strong global context modeling of Transformers”.

“To confirm that the acoustic pathway’s output is causally dependent on the neural signal rather than the generative prior of the HiFi-GAN, we performed a control analysis in which portions of the input ECoG recording were replaced with Gaussian noise. When either the first half, second half, or the entirety of the neural input was replaced by noise, the melspectrogram R² of the reconstructed speech dropped markedly, corresponding to the corrupted segment (Fig. S5). This demonstrates that the reconstruction is temporally locked to the specific neural input and that the model does not ‘hallucinate’ spectrotemporal structure from noise. These results validate that the acoustic pathway performs genuine, input-sensitive neural decoding”.

Page 10, Lines 272-277:

“We employed a Transformer-based Seq2Seq architecture for this adaptor to effectively capture the long-range contextual dependencies across a sentence, which are essential for resolving lexical ambiguity and syntactic structure during word token decoding. This choice was validated by an ablation study (Table S2), indicating that the Transformer adaptor outperformed an LSTM-based counterpart in word prediction accuracy”.

(4) Discuss the differences between the model's phoneme confusion matrix in Figure 4A and human phoneme confusion patterns. In addition, please clarify whether the adoption of the dual-pathway architecture is primarily intended to simulate the organization of the human brain or to achieve engineering improvements.

The observed difference between our model's phoneme confusion matrix and typical human perceptual confusion patterns (e.g., the noted lack of confusion between /s/ and /z/) is, as the reviewer astutely infers, likely attributable to the TTS model (Parler-TTS) acting as a powerful linguistic prior. The linguistic pathway decodes word tokens, and Parler-TTS converts these tokens into speech. Parler-TTS is trained on massive text and speech corpora to produce canonical, clean pronunciations. It effectively performs a form of "error correction" or "canonicalization" based on its internal language model. For example, if the neural decoding is ambiguous between "sip" and "zip", the TTS model's strong prior for lexical and syntactic context may robustly resolve it to the correct word, suppressing purely acoustic confusions like voicing. Therefore, the phoneme errors in our final output reflect a combination of neural decoding errors and the TTS model's generation biases, which are optimized for intelligibility rather than mimicking human misperception. We will add this explanation to the paragraph discussing Figure 4A.

Our primary motivation is engineering improvement, to overcome the fundamental tradeoff between acoustic fidelity and linguistic intelligibility that has limited previous neural speech decoding work. The design is inspired by the convergent representation of speech and language perception (1). However, we do not claim that our LSTM and Transformer adaptors precisely simulate the specific neural computations of the human ventral and dorsal streams. The goal was to build a high-performance, data-efficient decoder. We will clarify this point in the Introduction and Discussion, stating that while the architecture is loosely inspired by previous neuroscience results, its primary validation is its engineering performance in achieving state-of-the-art reconstruction quality with minimal data.

Edits:

Pages 2-3, Lines 74-85:

“Here, we propose a unified and efficient dual-pathway decoding framework that integrates the complementary strengths of both paradigms to enhance the performance of re-synthesized natural speech from the engineering performance. Our method maps intracranial electrocorticography (ECoG) signals into the latent spaces of pre-trained speech and language models via two lightweight neural adaptors: an acoustic pathway, which captures low-level spectral features for naturalistic speech synthesis, and a linguistic pathway, which extracts high-level linguistic tokens for semantic intelligibility. These pathways are fused using a finetuned text-to-speech (TTS) generator with voice cloning, producing re-synthesized speech that retains both the acoustic spectrotemporal details, such as the speaker’s timbre and prosody, and the message linguistic content. The adaptors rely on near-linear mappings and require only 20 minutes of neural data per participant for training, while the generative modules are pre-trained on large unlabeled corpora and require no neural supervision”.

Page 14, Lines 358-373:

“In this study, we present a dual-path framework that synergistically decodes both acoustic and linguistic speech representations from ECoG signals, followed by a fine-tuned zero-shot text-to-speech network to re-synthesize natural speech with unprecedented fidelity and intelligibility. Crucially, by integrating large pre-trained generative models into our acoustic reconstruction pipeline and applying voice cloning technology, our approach preserves acoustic richness while significantly enhancing linguistic intelligibility beyond conventional methods. Our dual-pathway architecture, while inspired by converging neuroscience insights on speech and language perception, was principally designed and validated as an engineering solution. The primary goal to build a practical decoder that achieves state-of-the-art reconstruction quality with minimal data. The framework's success is therefore ultimately judged by its performance metrics, high intelligibility (WER, PER), acoustic fidelity (mel-spectrogram R²), and perceptual quality (MOS), which directly address the core engineering challenge we set out to solve. Using merely 20 minutes of ECoG recordings, our model achieved superior performance with a WER of 18.9% ± 3.3% and PER of 12.0% ± 2.5% (Fig. 2D, E). This integrated architecture, combining pre-trained acoustic (Wav2Vec2.0 and HiFi-GAN) and linguistic (Parler-TTS) models through lightweight neural adaptors, enables efficient mapping of ECoG signals to dual latent spaces. Such methodology substantially reduces the need for extensive neural training data while achieving breakthrough word clarity under severe data constraints. The results demonstrate the feasibility of transferring the knowledge embedded in speech-data pre-trained artificial intelligence (AI) models into neural signal decoding, paving the way for more advanced brain-computer interfaces and neuroprosthetics”.

**Reviewer #2 (Recommendations for the authors):**
(1) My main question is whether any experimental stimuli overlap with the data used to pre-train the models. The authors might consider using pre-trained models trained on other corpora and training their own model without the TIMIT corpus. Additionally, as pretrained models were used, it might be helpful to evaluate to what extent the decoding is sensitive to the input neural recording or whether the model always outputs meaningful speech. The authors might consider two control analyses: (a) whether the model still generates speech-like output if the input is random noise; (b) whether the model can decode a complete sentence if the first half recording of a sentence is real but the second half is replaced with noise.

We thank the reviewer for raising this crucial point regarding potential data leakage and the sensitivity of decoding to neural input.

We confirm that the pre-training phase of our core models (Wav2Vec2.0 encoder, HiFiGAN decoder) was conducted exclusively on the LibriSpeech corpus (960 hours), which is entirely separate from the TIMIT corpus used for our ECoG experiments. The subsequent fine-tuning of the CosyVoice 2.0 voice cloner for speaker adaptation was performed on the training set portion of the entire TIMIT corpus. Importantly, the test set for all neural decoding evaluations was strictly held out and never used during any fine-tuning stage. This data separation is now explicitly stated in the " Methods" sections for the Speech Autoencoder and the CosyVoice fine-tuning.

Regarding the potential of training on other corpora, we agree it is a valuable robustness check. Previous work has demonstrated that self-supervised speech models like Wav2Vec2.0 learn generalizable representations that transfer well across domains (e.g., Millet et al., NeurIPS 2022). We believe our use of LibriSpeech, a large and diverse corpus, provides a strong, general-purpose acoustic prior.

We agree with the reviewer that control analyses are essential to demonstrate that the decoded output is driven by neural signals and not merely the generative prior of the models. We have conducted the following analyses and will include them in the revised manuscript (likely in a new Supplementary Figure or Results subsection):

(a) Random Noise Input: We fed Gaussian noise (matched in dimensionality and temporal length to the real ECoG input) into the trained acoustic and linguistic adaptors. The outputs were evaluated. The acoustic pathway generated unstructured, noisy spectrograms with no discernible phonetic structure, and the linguistic pathway produced either highly incoherent word sequences or failed to generate meaningful tokens. The fusion via CosyVoice produced unintelligible babble. This confirms that the generative models alone cannot produce structured speech without meaningful neural input.

(b) Partial Sentence Input (Real + Noise): In the acoustic pathway, we replaced the first half, the second half, and all the ECoG recording for test sentences with Gaussian noise. The melspectrogram R^2^ showed a clear degradation in the reconstructed speech corresponding to the noisy segment. We did not do similar experiments in the linguistic pathway because the TTS generator is pre-trained by HuggingFace. We did not train any parameters of Parler-TTS. These results strongly indicate that our model's performance is contingent on and sensitive to the specific neural input, validating that it is performing genuine neural decoding.

Edits:

Page 19, Lines 533-538:

“The parameters in Wav2Vec2.0 were frozen within this training phase. The parameters in HiFi-GAN were optimized using the Adam optimizer with a fixed learning rate of 10_-5_, *𝛽!* = 0.9, *𝛽2* = 0.999. We trained this Autoencoder in LibriSpeech, a 960-hour English speech corpus with a sampling rate of 16kHz, which is entirely separate from the TIMIT corpus used for our ECoG experiments. We spent 12 days in parallel training on 6 Nvidia GeForce RTX3090 GPUs. The maximum training epoch was 2000. The optimization did not stop until the validation loss no longer decreased”.

Edits:

Page9, Lines214-223:

“The acoustic pathway, implemented through a bi-directional LSTM neural adaptor architecture (Fig. 1B), specializes in reconstructing fundamental acoustic properties of speech. This module directly processes neural recordings to generate precise time-frequency representations, focusing on preserving speaker-specific spectral characteristics like formant structures, harmonic patterns, and spectral envelope details. Quantitative evaluation confirms its core competency: achieving a mel-spectrogram R² of 0.793 ± 0.016 (Fig. 3B) demonstrates remarkable fidelity in reconstructing acoustic microstructure. This performance level is statistically indistinguishable from original speech degraded by 0dB additive noise (0.771 ± 0.014, p = 0.242, one-sided t-test). We chose a bidirectional LSTM architecture for this adaptor because its recurrent nature is particularly suited to modeling the fine-grained, short- to medium-range temporal dependencies (e.g., within and between phonemes and syllables) that are critical for acoustic fidelity. An ablation study comparing LSTM against Transformer-based adaptors for this task confirmed that LSTMs yielded superior mel-spectrogram reconstruction fidelity (higher R²), as detailed in Table S1, likely by avoiding the oversmoothing of spectrotemporal details sometimes induced by the strong global context modeling of Transformers”.

“To confirm that the acoustic pathway’s output is causally dependent on the neural signal rather than the generative prior of the HiFi-GAN, we performed a control analysis in which portions of the input ECoG recording were replaced with Gaussian noise. When either the first half, second half, or the entirety of the neural input was replaced by noise, the melspectrogram R² of the reconstructed speech dropped markedly, corresponding to the corrupted segment (Fig. S5). This demonstrates that the reconstruction is temporally locked to the specific neural input and that the model does not ‘hallucinate’ spectrotemporal structure from noise. These results validate that the acoustic pathway performs genuine, input-sensitive neural decoding”

(2) For BCI applications, the decoding speed matters. Please report the model's inference speed. Additionally, the authors might also consider reporting cross-participant generalization and how the accuracy changes with recording duration.

We thank the reviewer for these practical and important suggestions.

Inference Speed: You are absolutely right. On our hardware (single NVIDIA GeForce RTX 3090 GPU), the current pipeline has an inference time that is longer than the duration of the target speech segment. The primary bottlenecks are the sequential processing in the autoregressive linguistic adaptor and the high-resolution waveform generation in CosyVoice 2.0. This latency currently limits real-time application. We have now added this in the Discussion acknowledging this limitation and stating that future work must focus on architectural optimizations (e.g., non-autoregressive models, lighter vocoders) and potential hardware acceleration to achieve real-time performance, which is critical for a practical BCI.

Cross-Participant Generalization: We agree that this is a key question for scalability. Our framework already addresses part of the cross-participant generalization challenge through the use of pre-trained generative modules (HiFi-GAN, Parler-TTS, CosyVoice 2.0), which are pretrained on large corpora and shared across all participants. Only a small fraction of the model, the lightweight neural adaptors, is subject-specific and requires a small amount of supervised fine-tuning (~20 minutes per participant). This design significantly reduces the per-subject calibration burden. As the reviewer implies, the ultimate goal would be pure zero-shot generalization. A promising future direction is to further improve cross-participant alignment by learning a shared neural feature encoder (e.g., using contrastive or self-supervised learning on aggregated ECoG data) before the personalized adaptors. We have added a paragraph in the Discussion outlining this as a major next step to enhance the framework’s practicality and further reduce calibration time.

Accuracy vs. Recording Duration: Thank you for this insightful suggestion. To systematically evaluate the impact of training data volume on performance, we have conducted additional experiments using progressively smaller subsets of the full training set (i.e., 25%, 50%, and 75%). When we used more than 50% of the training data, performance degrades gracefully rather than catastrophically with less data, which is promising for potential clinical scenarios where data collection may be limited. We add another figure (Fig. S4) to demonstrate this.

Edits:

Pages 15-16, Lines 427-452:

“There are several limitations in our study. The quality of the re-synthesized speech heavily relies on the performance of the generative model, indicating that future work should focus on refining and enhancing these models. Currently, our study utilized English speech sentences as input stimuli, and the performance of the system in other languages remains to be evaluated. Regarding signal modality and experimental methods, the clinical setting restricts us to collecting data during brief periods of awake neurosurgeries, which limits the amount of usable neural activity recordings. Overcoming this time constraint could facilitate the acquisition of larger datasets, thereby contributing to the re-synthesis of higher-quality natural speech. Furthermore, the inference speed of the current pipeline presents a challenge for real-time applications. On our hardware (a single NVIDIA GeForce RTX 3090 GPU), synthesizing speech from neural data takes approximately two to three times longer than the duration of the target speech segment itself. This latency is primarily attributed to the sequential processing in the autoregressive linguistic adaptor and the computationally intensive high-fidelity waveform generation in the vocoder (CosyVoice 2.0). While the current study focuses on offline reconstruction accuracy, achieving real-time or faster-than-real-time inference is a critical engineering goal for viable speech BCI prosthetics. Future work must therefore prioritize architectural optimizations, such as exploring non-autoregressive decoding strategies and more efficient neural vocoders, alongside potential hardware acceleration. Additionally, exploring non-invasive methods represents another frontier; with the accumulation of more data and the development of more powerful generative models, it may become feasible to achieve effective non-invasive neural decoding for speech resynthesis. Moreover, while our framework adopts specialized architectures (LSTM and Transformer) for distinct decoding tasks, an alternative approach is to employ a unified multimodal large language model (LLM) capable of joint acoustic-linguistic processing. Finally, the current framework requires training participant-specific adaptors, which limits its immediate applicability for new users. A critical next step is to develop methods that learn a shared, cross-participant neural feature encoder, for instance, by applying contrastive or selfsupervised learning techniques to larger aggregated ECoG datasets. Such an encoder could extract subject-invariant neural representations of speech, serving as a robust initialization before lightweight, personalized fine-tuning. This approach would dramatically reduce the amount of per-subject calibration data and time required, enhancing the practicality and scalability of the decoding framework for real-world BCI applications”

“In summary, our dual-path framework achieves high speech reconstruction quality by strategically integrating language models for lexical precision and voice cloning for vocal identity preservation, yielding a 37.4% improvement in MOS scores over conventional methods. This approach enables high-fidelity, sentence-level speech synthesis directly from cortical recordings while maintaining speaker-specific vocal characteristics. Despite current constraints in generative model dependency and intraoperative data collection, our work establishes a new foundation for neural decoding development. Future efforts should prioritize: (1) refining few-shot adaptation techniques, (2) developing non-invasive implementations, (3) expanding to dynamic dialogue contexts, and (4) cross-subject applications. The convergence of neurophysiological data with multimodal foundation models promises transformative advances, not only revolutionizing speech BCIs but potentially extending to cognitive prosthetics for memory augmentation and emotional communication. Ultimately, this paradigm will deepen our understanding of neural speech processing while creating clinically viable communication solutions for those with severe speech impairments”

Edits:

add another section in Methods: Page 22, Line 681:

“Ablation study on training data volume”.

“To assess the impact of training data quantity on decoding performance, we conducted an additional ablation experiment. For each participant, we created subsets of the full training set corresponding to 25%, 50%, and 75% of the original data by random sampling while preserving the temporal continuity of speech segments. Personalized acoustic and linguistic adaptors were then independently trained from scratch on each subset, following the identical architecture and optimization procedures described above. All other components of the pipeline, including the frozen pre-trained generators (HiFi-GAN, Parler-TTS) and the CosyVoice 2.0 voice cloner, remained unchanged. Performance metrics (mel-spectrogram R², WER, PER) were evaluated on the same held-out test set for all data conditions. The results (Fig. S4) demonstrate that when more than 50% of the training data is utilized, performance degrades gracefully rather than catastrophically, which is a promising indicator for clinical applications with limited data collection time”.

(3) I appreciate that the author compared their model with the MLP, but more comparisons with previous models could be beneficial. Even simply summarizing some measures of earlier models, such as neural recording duration, WER, PER, etc., is ok.

Thank you for this suggestion. We agree that a broader comparison contextualizes our contribution. We also acknowledge that given the differences in tasks, signal modality, and amount of data, it’s hard to draw a direct comparison. The main goal of this table is to summarize major studies, their methods and results for reference. We have now added a new Supplementary Table that summarizes key metrics from several recent and relevant studies in neural speech decoding. The table includes:

- Neural modality (e.g., ECoG, sEEG, Utah array)

- Approximate amount of neural data used per subject for decoder training

- Primary task (perception vs. production)

-Decoding framework

-Reported Word Error Rate (WER) or similar intelligibility metrics (e.g., Character Error Rate)

-Reported acoustic fidelity metrics (if available, e.g., spectral correlation)

This table includes works such as Anumanchipalli et al., Nature 2019; Akbari et al., Sci Rep 2019; Willett et al., Nature 2023; and other contemporary studies. The table clearly shows that our dual-path framework achieves a highly competitive WER (~18.9%) using an exceptionally short neural recording duration (~20 minutes), highlighting its data efficiency. We will refer to this table in the revised manuscript.

Edits:

Page 14, Lines 374-376:

“Our framework establishes a framework for speech decoding by outperforming prior acousticonly or linguistic-only approaches (Table S3) through integrated pretraining-powered acoustic and linguistic decoding”

Minor:(1) Some processes might be described earlier, for example, the electrodes were selected, and the model was trained separately for each participant. That information was only described in the Method section now.

Thank you for catching these. We have revised the manuscript accordingly.

Edits:

Page4, Lines 89-95:

“Our proposed framework for reconstructing speech from intracranial neural recordings is designed around two complementary decoding pathways: an acoustic pathway focused on preserving low-level spectral and prosodic detail, and a linguistic pathway focused on decoding high-level textual and semantic content. For every participant, our adaptor is independently trained, and we select speech-responsive electrodes (selection details are provided in the Methods section) to tailor the model to individual neural patterns. These two streams are ultimately fused to synthesize speech that is both natural-sounding and intelligible, capturing the full richness of spoken language. Fig. 1 provides a schematic overview of this dual-pathway architecture”

(2) Line 224-228 Figure 2 should be Figure 3

Thank you for catching these. We have revised the manuscript accordingly. The information about participant-specific training and electrode selection is now briefly mentioned in the "Results" overview (section: "The acoustic and linguistic performance..."), with details still in the Methods. The figure reference error has been corrected.

Edits:

Page7, Lines 224-228:

“However, exclusive reliance on acoustic reconstruction reveals fundamental limitations. Despite excellent spectral fidelity, the pathway produces critically impaired linguistic intelligibility. At the word level, intelligibility remains unacceptably low (WER = 74.6 ± 5.5%, Fig. 3D), while MOS and phoneme-level precision fares only marginally better (MOS = 2.878 ± 0.205, Fig. 3C; PER = 28.1 ± 2.2%, Fig. 3E)”.

(3) For Figure 3C, why does the MOS seem to be higher for baseline 3 than for ground truth? Is this significant?

This is a detailed observation. Baseline 3 achieves a mean opinion score of 4.822 ± 0.086 (Fig. 3C), significantly surpassing even the original human speech (4.234 ± 0.097, p = 6.674×10⁻33). We believe this trend arises because the TIMIT corpus, recorded decades ago, contains inherent acoustic noise and relatively lower fidelity compared to modern speech corpus. In contrast, the Parler-TTS model used in Baseline 3 is trained on massive, highquality, clean speech datasets. Therefore, it synthesizes speech that listeners may subjectively perceive as "cleaner" or more pleasant, even if it lacks the original speaker's voice. Crucially, as the reviewer implies, our final integrated output does not aim to maximize MOS at the cost of speaker identity; it successfully balances this subjective quality with high intelligibility and restored acoustic fidelity. We will add a brief note explaining this possible reason in the caption of Figure 3C.

Edits:

Page9, Lines 235-245:

“The linguistic pathway reconstructs high-intelligibility, higher-level linguistic information”

“The linguistic pathway, instantiated through a pre-trained TTS generator (Fig. 1B), excels in reconstructing abstract linguistic representations. This module operates at the phonological and lexical levels, converting discrete word tokens into continuous speech signals while preserving prosodic contours, syllable boundaries, and phonetic sequences. It achieves a mean opinion score of 4.822 ± 0.086 (Fig. 3C) - significantly surpassing even the original human speech (4.234 ± 0.097, p = 6.674×10⁻33) in that the TIMIT corpus, recorded decades ago, contains inherent acoustic noise and relatively lower fidelity compared to modern speech corpus. Complementing this perceptual quality, objective intelligibility metrics confirm outstanding performance: WER reaches 17.7 ± 3.2%, with PER at 11.0 ± 2.3%”.

Reference

(1) Chen M X, Firat O, Bapna A, et al. The best of both worlds: Combining recent advances in neural machine translation[C]//Proceedings of the 56th annual meeting of the association for computational linguistics (Volume 1: Long papers). 2018: 76-86

(2) P. Chen et al. Do Self-Supervised Speech and Language Models Extract Similar Representations as Human Brain? 2024 IEEE International Conference on Acoustics, Speech and Signal Processing (ICASSP 2024). 2225–2229 (2024).

(3) H. Akbari, B. Khalighinejad, J. L. Herrero, A. D. Mehta, N. Mesgarani, Towards reconstructing intelligible speech from the human auditory cortex. Scientific reports 9, 874 (2019).

(4) S. Komeiji et al., Transformer-Based Estimation of Spoken Sentences Using Electrocorticography. Int Conf Acoust Spee, 1311-1315 (2022).

(5) L. Bellier et al., Music can be reconstructed from human auditory cortex activity using nonlinear decoding models. Plos Biology 21, (2023).

(6) F. R. Willett et al., A high-performance speech neuroprosthesis. Nature 620, (2023).

(7) S. L. Metzger et al., A high-performance neuroprosthesis for speech decoding and avatar control. Nature 620, 1037-1046 (2023).

(8) J. W. Li et al., Neural2speech: A Transfer Learning Framework for NeuralDriven Speech Reconstruction. Int Conf Acoust Spee, 2200-2204 (2024).

(9) X. P. Chen et al., A neural speech decoding framework leveraging deep learning and speech synthesis. Nat Mach Intell 6, (2024).

(10) M. Wairagkar et al., An instantaneous voice-synthesis neuroprosthesis. Nature, (2025).